# PROGRAMMABLE SYNTHETIC DATA GENERATION

## ABSTRACT

Large amounts of tabular data remain underutilized due to privacy, data quality, and data sharing limitations. While generating synthetic data resembling the original distribution addresses some of these issues, most applications would benefit from additional customization on the generated data. However, existing synthetic data generation approaches are limited to particular constraints, *e.g.,* differential privacy (DP) or fairness. In this work, we introduce ProgSyn, the first programmable and flexible synthetic tabular data generation framework. Customization is achieved via programmatically declared statistical and logical expressions, supporting a wide range of requirements (*e.g.,* DP or fairness, among others). To ensure high synthetic data quality in the presence of custom specifications, ProgSyn pre-trains a generative model on the original dataset and fine-tunes it on a differentiable loss automatically derived from the provided specifications using novel relaxations. We conduct an extensive experimental evaluation of ProgSyn over four datasets and on numerous custom specifications, where we outperform state-of-the-art specialized approaches on several tasks while being more general. For instance, at the same fairness level, we achieve 2.3% higher downstream accuracy than the state-of-the-art in fair synthetic data generation on the Adult dataset.

## 1 INTRODUCTION

The availability of large datasets has been key to the rapid progress of machine learning. To enable this progress, datasets often have to be shared between different organizations and potentially passed on to third parties to train machine learning models. This often presents a roadblock as data owners are responsible for ensuring they do not perpetuate biases present in the data and do not violate user privacy by sharing their personal records. Tabular data is especially delicate from this perspective, as it is abundant in high-stakes applications, such as finance and healthcare (Borisov et al., 2022). An emerging and promising approach for addressing these issues is synthetic data generation.

**Synthetic data** The promise of synthetic data is to produce a new dataset statistically resembling the original while overcoming the above issues. Driven by recent regulations requiring bias mitigation (*e.g.,* GDPR European Parliament & Council of the European Union (2016) Art. 5a), data accuracy (GDPR Art. 5d), and privacy (GDPR Art. 5c and 5e), there has been increased interest in this field.

Prior work has only addressed some of the data sharing concerns: differentially private synthetic data (*e.g.,* Zhang et al. (2014); Jordon et al. (2019); McKenna et al. (2022)), generating data with reduced bias (*e.g.,* van Breugel et al. (2021); Rajabi & Garibay (2022)), and combining these two objectives (Pujol et al., 2022). However, these methods might still generate data violating truthfulness (*e.g.,* person who is 10 years old and has a doctorate) or containing undesired statistical patterns (*e.g.,* a pharmaceutical company not sharing even synthetic copies of their clinical trial data, as the distribution of patient conditions reveals their development focus). Therefore, it remains a key challenge to enable data owners to generate custom high-utility data as required by their applications.

**ProgSyn: Programmable Synthetic Data Generation** In this work, we introduce the first synthetic tabular data generation method allowing for general programmable customization. Figure 1 shows an overview of ProgSyn, featuring example specifications defined by the data owner, where no person younger than 25 with a doctorate should be generated and where bias w.r.t sex should be minimized.

ProgSyn supports a wide range of customizations. First, it allows for differentially private training protecting individuals included in the original dataset. Through logical and implication constraints

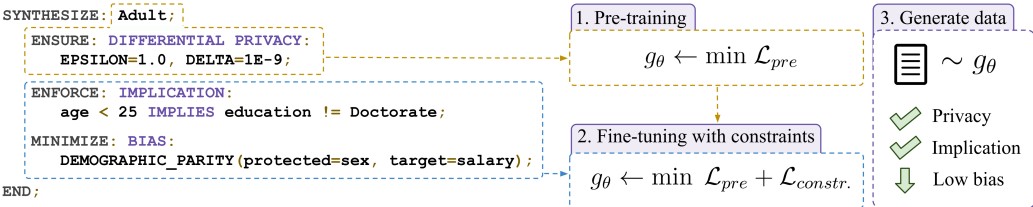

Figure 1: An overview of ProgSyn. The data owner writes a program that lists specifications for the synthetic data. For example, they might want to make sure that the model does not generate people younger than 25 with a Doctorate degree. Additionally, they might require that the synthetic data is differentially private and unbiased. To achieve this, ProgSyn pre-trains a differentially private generative model, and then fine-tunes it to adhere to the given specifications. Finally, the generative model can be used to sample a synthetic dataset with the desired properties.

it can specify relationships that each data point has to satisfy (as in Figure 1). Through statistical specifications, it allows users to directly manipulate statistical properties of the synthetic data. Finally, it provides soft-constraints for encouraging desirable behavior of classifiers trained on the synthetic data (*e.g.,* low bias). Thus, ProgSyn generalizes prior works supporting only restricted specifications.

Our key insight is that one can preserve high utility by pre-training a generative model ($g_\theta$ in Figure 1) on the original dataset and then fine-tune it to fit custom specifications. ProgSyn automatically converts the non-differentiable specifications into a relaxed differentiable loss which is then minimized together with the pre-training objective, biasing the model towards the desired custom distribution.

**Example: Statistical Manipulations** We demonstrate on a practical example how statistical manipulations allowed by ProgSyn enable an organization to share their synthesized data without compromising proprietary information. Recall that a drug company may need to obfuscate the distribution of patient conditions before even sharing synthetic data, as they need to avoid revealing the focus of their research. We instantiate this example on the Health Heritage dataset, containing patient data. As shown in Figure 2, specifying ProgSyn to increase the feature's entropy, it obfuscates the details of patient conditions, making it

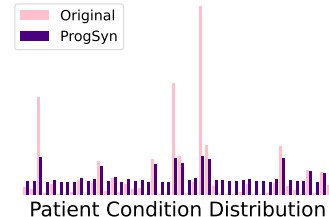

Figure 2: ProgSyn obfuscating the distribution using statistical manipulations

difficult to accurately determine the exact prevalence of the most common conditions. Meanwhile, it retains high quality in the synthetic data only losing $1\%$ downstream accuracy w.r.t. the original data.

In our experimental evaluation, we demonstrate that ProgSyn produces synthetic data according to a number of custom specifications unsupported by prior work, while achieving high utility. Furthermore, on specifications supported by prior work we either outperform them or at least match their performance. For instance, we improve the state-of-the-art in fair synthetic data generation on the Adult (Dua & Graff, 2017) dataset by achieving a $2.3\%$ higher downstream accuracy and a $2\times$ lower demographic parity distance of $0.01$. Additionally, we demonstrate that ProgSyn is able to stack several diverse specifications at the same time, while maintaining high data quality.

**Main contributions** Our key contributions are:

1. The first framework for programmable synthetic tabular data generation, supporting a wide range of customizations on the generated data.

2. Novel relaxations allowing for fine-tuning a generative model via differentiable regularizers derived from the specifications, while retaining high synthetic data quality.

3. An implementation of the framework in a system called ProgSyn, together with an extensive evaluation demonstrating its strong competitiveness and versatility.

## 2 BACKGROUND

**Tabular Data** Tabular data is one of the most common data formats, extensively used in high-stakes contexts, *e.g.,* in healthcare, finance, and social sciences (Borisov et al., 2022). We assume that the data at hand *only* contains discrete columns, *i.e.,* we discretize any continuous columns before proceeding. Let the number of columns be $K$, then we denote the domain of each resulting discrete feature as $\mathcal{D}_i$ for $i \in [K]$. We one-hot encode the columns, turning each $d_i \in \mathcal{D}_i$ into a binary vector of length $|\mathcal{D}_i|$, with a single non-zero entry marking the position of the encoded discrete category. The resulting set of one-hot encoded rows is denoted as $\mathcal{X}$, where each encoded data point $x \in \mathcal{X}$ is of length $q := \sum_{i=1}^{K} |\mathcal{D}_i|$ and contains exactly $K$ non-zero entries. Further, a full table of $N$ rows is denoted as $X \in \mathcal{X}^N$, with $X_i$ denoting the $i$-th data point. In the rest of this text, we will also refer to $X$ as a sample of size $N$, as well as simply a dataset, and will use row and data point interchangeably to refer to a single $x \in X$. Also, unless stated otherwise, we will denote a synthetic sample as $\hat{X}$. Finally, let $\mathcal{S} := \{s_1, \dots, s_m\} \subseteq [K] := \{1, \dots, K\}$, then we write $X[\mathcal{D}_{s_1}, \dots, \mathcal{D}_{s_m}]$ meaning only the (column-space) subset of $X$ that corresponds to the original columns $\mathcal{D}_{s_1}, \dots, \mathcal{D}_{s_m}$.

**Marginals** Let $\mathcal{S} := \{\mathcal{D}_{s_i}\}_{i=1}^{m}$ be a subset of $m$ columns, then the $m$-way marginal over $\mathcal{S}$ on a sample $X$ counts the occurrences of each feature combination from the product space $\bigtimes_{i=1}^{m} \mathcal{D}_{s_i}$ over all rows in $X$. We denote the unnormalized marginal as $\mu(\mathcal{S}, X)$, and denote the normalized marginal as $\bar{\mu}(\mathcal{S}, X) := \frac{1}{N} \mu(\mathcal{S}, X)$. Marginals are an important statistic in tabular data, as they effectively capture the approximate distributional characteristics of the features in the sample, facilitating the calculation of a wide range of statistics, *e.g.,* correlations and conditional relationships between the involved features. Additionally, due to the one-hot encoding in $X$, we can differentiably calculate marginals using the Kronecker product, *e.g., :* $\mu(\mathcal{S}, X) := \sum_{k=1}^{N} X_k[\mathcal{D}_{s_1}] \otimes \dots \otimes X_k[\mathcal{D}_{s_m}]$.

**Differential Privacy** The gold standard for providing privacy guarantees for data dependent algorithms is differential privacy (DP) (Dwork, 2006), where the privacy of individual contained in a dataset is ensured by limiting the impact a single data point can have on the outcome of the algorithm. This is usually achieved by injecting carefully engineered noise in the process, which in turn negatively affects the accuracy of the procedure. The privacy level is quantified by $\epsilon$, with lower levels of $\epsilon$ corresponding to higher privacy, and as such, higher noise and lower accuracy.

**Fair Classification** As machine learning systems may propagate biases from their training data (Corbett-Davies et al., 2017; Buolamwini & Gebru, 2018), there is an increased interest to mitigate this effect (Dwork et al., 2012; Benaich & Hogarth, 2021; Chiu et al., 2021). The demographic parity distance is a fairness measure quantifiyng the difference in expected outcomes based on an individual's protected group membership $\mathcal{D}_s$: $\pi_{\mathcal{D}_s}(f, \mathcal{X}) := \max_{d_i, d_j \in \mathcal{D}_s \times \mathcal{D}_s} |E_{x \sim \mathcal{X}}[f(x)|\mathcal{D}_s = d_i] - E_{x \sim \mathcal{X}}[f(x)|\mathcal{D}_s = d_j]|$, where $f$ is a classifier.

**Synthetic Data** The goal of synthetic data generation is to train a generative model $g_\theta$ on the real data $X$ to produce synthetic samples $\hat{X}$ that are statistically as close as possible to $X$. Ultimately, $\hat{X}$ should have high enough quality to replace $X$ in data analysis and machine learning tasks.

## 3 RELATED WORK

**Synthetic Tabular Data: Nominal Approaches** Unconstrained, or nominal synthetic tabular data generation exhibits a long line of work, where the most prominent approaches are collected in the Synthetic Data Vault (SDV) (Patki et al., 2016), including the deep learning based methods of TVAE and CTGAN Xu et al. (2018). Although recent works (Liu et al., 2023; Kotelnikov et al., 2022; Borisov et al., 2023) improved over the models in SDV, they lack an extensive support for, privacy, fairness, or other customizations. Our work is the first general approach in this direction.

**Differentially Private Synthetic Data** As nominal synthetic data does not provide sufficient privacy protection (Stadler et al., 2022), differentially private (DP) synthetic data is of increasing interest. A recent survey (Tao et al., 2021) established that generative adversarial networks (GAN) (*e.g.,* PATE-GAN (Jordon et al., 2019) and DP-CGAN (Torkzadehmahani et al., 2019)) are outperformed

by marginal-based graphical models operating on a fixed set of measurements (*e.g.,* PrivBayes (Zhang et al., 2014), and MST (McKenna et al., 2022)). Recently, algorithms that iteratively make new measurements of target marginals have shown strong improvements (*e.g.,* RAP (Aydöre et al., 2021), GEM (Liu et al., 2021), and AIM (McKenna et al., 2021)); but still lack extensive customizability.

**Fair Synthetic Tabular Data**   Reducing bias of synthetic data is an important concern, especially under DP, where the effecs of bias are exacerbated (Ganev et al., 2022). Most works in this area make use of GANs with bias-penalized loss functions to encourage fairness (Xu et al., 2019b;a; Abroshan et al., 2022; Rajabi & Garibay, 2022), or pre-processing the dataset to remove bias before training a generative model Chaudhari et al. (2022). In a different approach, DECAF (van Breugel et al., 2021) trains a causally-aware GAN, and removes undesired causal relationships at generation time reducing bias. Finally, PreFair (Pujol et al., 2022) extends the graphical model based DP algorithm of McKenna et al. (2021), reducing bias by prohibiting undesired connections in the underlying graph.

**Synthetic Tabular Data with Logical Constraints**   Although it is important for to allow for enforcing logical relationships in the synthetic data, only few works have considered this issue. AIM (McKenna et al., 2022) allows a restricted set of constraints by manually zeroing out certain entries in the marginals. As we find in Section 5, this approach can severely impact the quality of the generated data. Kamino (Ge et al., 2020) is a DP synthetic tabular data generation method focused on facilitating logical relationships between pairs of generated data points. As ProgSyn operates under the usual assumption of i.i.d. data, the constraints supported by Kamino do not extend to our setting.

**Constraints in Continuous Models**   There has been a long line of work focusing on encoding domain-knowledge or other information in the form of *logical constraints* that would aid the machine learning model in its performance. Some prominent works achieve this by modifying the loss function or its computation at training time (Manhaeve et al., 2018; Fischer et al., 2019; Nandwani et al., 2019; Rajaby Faghihi et al., 2021; Yang et al., 2022; Li et al., 2023), or by modifying the model and/or its inference procedure (Hu et al., 2016; Hoernle et al., 2022; Ahmed et al., 2022; Badreddine et al., 2022). The main distinguishing factors with our work are: (i) these approaches improve the trained models by injecting additional knowledge, while ProgSyn's aim is freely customizable synthetic data generation; and (ii) most such approaches only support customizations that are limited to first order logic on individual data points, while ProgSyn also supports statistical and downstream customizations on the generated dataset. Further, note that ProgSyn's contribution is *not* another general language for differentiable logic, but one specific to synthetic data generation.

## 4   PROGSYN: PROGRAMMABLE SYNTHETIC DATA GENERATOR

To fully utilize tabular data, it is often necessary to create a customized synthetic version, beyond a nominal synthetic copy. Consider protecting individual privacy using differential privacy, supporting logical constraints to preserve or inject structure, allowing to directly influence statistical relationships, and facilitating classifiers trained on the synthetic data with desirable properties, *e.g.,* low bias in addition to high accuracy. While prior work considered small disjunct subsets of such customizations, ProgSyn is the first method that allows for joint programmable specification of *all* of the above for synthetic data generation. ProgSyn converts the user-provided specifications to a differentiable loss and uses it to train the generative model. We now describe the underlying generative model, training procedure, and the technical details of the supported specifications.

### 4.1   THE PROGSYN FRAMEWORK

Following Liu et al. (2021), in the base generative model, we make use of the generator of a GAN to generate datasets from random noise, which is then trained by comparing marginals calculated on this generated dataset to the marginals of the original dataset. Formally, denote the parametric generative model as $g_\theta$, then $g_\theta : \mathbb{R}^p \to \mathcal{X}$ is a mapping to the one-hot representation-space of the original dataset. The input to $g_\theta$ is standard normal random noise $z$, *i.e.,* $z \sim \mathcal{N}(0, \mathbb{I}_{p \times p})$ (shorthand: $\mathcal{N}_p$). As such, we can sample from $g_\theta$ by first sampling an input noise and feeding it through the network to obtain the corresponding dataset sample. To ensure that the output of $g_\theta$ is in the correct binary representation described in Section 2, we use a per-feature straight-through gumbel-softmax estimator (Jang et al., 2017) as the final layer, which differentiably produces one-hot representations

for each output feature. The goal of training is to match the distribution induced by $g_\theta$ to that of the original data, *i.e.,* to find a $\theta$ such that $P_{g_\theta} \approx P_x$.

**Non-Private Pre-Training**  For the non-private training of $g_\theta$, we first measure a set of marginals on the original dataset $X$, denoted as $M(X)$. To obtain the training loss $\mathcal{L}_M$, we calculate the total variation (TV) distance between the true marginals $M(X)$ and the marginals measured on a generated sample $M(g_\theta(z))$ of size $B$, *i.e.,* $\mathcal{L}_M(g_\theta(z), X) \coloneqq \frac{1}{2} |M(X) - M(g_\theta(z))|$, where $z \sim \mathcal{N}_p^B$. We then use iterative gradient-based optimization to minimize $\mathcal{L}_M$, resampling $z$ at each iteration.

**Differentially Private Pre-Training**  For DP training, we adapt the DP iterative framework of McKenna et al. (2022), exchanging the original graphical model with our $g_\theta$. Crucially, we also modify the budget adaptation step; in a similar vein to adaptive ODE solvers, we allow both for increasing and decreasing the per iteration DP budget, depending on the improvements observed in the previous step. For more details, we refer the reader to Appendix E.

**Training ProgSyn**  Depending on whether DP is a requirement, we first pre-train ProgSyn either by the non-private or the DP training method described above *without* any other specifications. Once pre-training is done, we fine-tune ProgSyn minimizing the pre-training objective $\mathcal{L}_M$ regularized by additional soft-constraints $\mathcal{L}_{\text{spec.}}^{(i)}$ derived from the provided $n$ specifications, resulting in the objective:

$$\mathcal{L}_{\text{fine}}(g_\theta(z), X, X_r) \coloneqq \mathcal{L}_M(g_\theta(z), X) + \sum_{i=1}^{n} \lambda_i \, \mathcal{L}_{\text{spec.}}^{(i)}(g_\theta(z), X_r), \tag{1}$$

where $\{\lambda_i\}_{i=1}^n$ are real valued parameters weighing the soft-constraints' impact on the objective, $X$ is the original dataset, and $X_r$ is a reference dataset, which is either the original dataset itself, or, to respect DP, a sample generated at the end of fine-tuning. The goal is to find a $\theta^*$ that minimizes the fine-tuning loss $\mathcal{L}_{\text{fine}}(g_\theta(z), X, X_r)$. We discuss the choice of $\{\lambda_i\}_{i=1}^n$ in Appendix D.1.

## 4.2 Privacy, Logical, Statistical, and Downstream Specifications

Using the ProgSyn program on the Adult dataset (Dua & Graff, 2017) shown in Figure 3 as a running example, we introduce the technical details of each supported specification below.

**ProgSyn Programs**  Each program begins with a command fixing the source dataset we wish to make a synthetic copy of and ends in an `END;` command. In between, we may specify all customizations over the learned synthetic distribution. If no specifications are given, $g_\theta$ is trained to maximally match the original dataset in a non-private manner. Each command consists of (i) an action description, defining how the optimizer should treat the resulting regularizer (maximize, minimize, enforce, or ensure); (ii) a command type description; (iii) an optional `PARAM` specification, setting the corresponding regularization weight $\lambda$, and (iv) an expression describing the specification directly in terms of the data attributes.

```
1. SYNTHESIZE: Adult;

2.   ENSURE: DIFFERENTIAL PRIVACY:
       EPSILON=1.0, DELTA=1E-9;

3.   ENFORCE: ROW CONSTRAINT:
       age > 35 AND age < 55;

4.   MINIMIZE: IMPLICATION:
       marital_status in {Divorced, Never_married}
         IMPLIES
       relationship not in {Husband, Wife};

5.   ENFORCE: STATISTICAL:
       E[age|sex=Male] == E[age|sex=Female];

6.   MINIMIZE: BIAS: PARAM 0.01:
       DEMOGRAPHIC_PARITY(protected=sex, target=salary);

7.   MINIMIZE: DOWNSTREAM: PARAM 0.05:
       DOWNSTREAM_ACCURACY(features=all, target=sex);

8. END;
```

Figure 3: ProgSyn program on the Adult dataset containing example commands for each supported constraint type.

**Differential Privacy Constraint**  ProgSyn can protect the privacy of individuals in $X$ with DP by using the constraint shown in line 2 of Figure 3. This ensures that the pre-training of $g_\theta$ is done by the iterative DP method described in Section 4.1, and that fine-tuning does not access the original dataset $X$. This constraint *guarantees* that ProgSyn respects DP at the given $\epsilon$ privacy level.

**Logical Constraints**    To avoid generating unrealistic data points or when aiming to incorporate domain knowledge, it is necessary to support logical constraints over individual rows. For instance, consider the constraint (denoted as $\phi$) in line 3 of Figure 3, requiring that each generated individual's age is between 35 and 55. We refer to such first order logical expressions consisting of feature-constant comparisons chained by logical `AND` and `OR` operations that have to hold for *each row* of the synthetic samples as *row constraints*. Returning to our example, $\phi$ consists of two comparisons $t_1 := \texttt{age > 35}$ and $t_2 := \texttt{age < 55}$. To enforce $\phi$ over $g_\theta$, we first negate the expression $\phi$ to obtain $\neg\phi = \texttt{age <= 35 OR age >= 55}$, and at each iteration count the rows where the negated expression holds, penalizing the fine-tuning loss with this count. However, as both hard logic and counting are non-differentiable, enforcing such constraints over the synthetic data is challenging. To circumvent this issue, we introduce a novel differentiable computation of a binary mask $b_{\neg\phi}$ marking the rows in a generated synthetic sample $\hat{X}$ of length $N$ that satisfy $\neg\phi$, the sum of which is exactly the number of rows violating $\phi$. We do this by making use of the differentiable one-hot encoding in $\hat{X}$. First, we translate each of the negated comparison terms $\neg t_1$ and $\neg t_2$ into binary masks $m_{\neg t_1}, m_{\neg t_2} \in \{0, 1\}^q$ over the columns by setting each coordinate that corresponds to a valid assignment in $t_i$ to 1 and keeping the rest of the entries 0. For instance, if the `age` feature is discretized as [18-35, 36-45, 46-54, 55-80], then $\neg t_1[\texttt{age}] = [1, 0, 0, 0]$ and $\neg t_2[\texttt{age}] = [0, 0, 0, 1]$, with the rest of the $q - 4$ dimensions padded with zeros. Finally, to compute the resulting binary mask $b_{\neg\phi}$ over the rows of $\hat{X}$, we introduce the following differentiable logical operator primitives: `AND`: $\hat{X}m_{t_1}^T \odot \hat{X}m_{t_2}^T$, and `OR`: $\hat{X}m_{t_1}^T + \hat{X}m_{t_2}^T - \hat{X}m_{t_1}^T \odot \hat{X}m_{t_2}^T$. In the case of composite logical expressions, we apply these primitives to each pair of comparisons recursively. Notice that as we only make use of matrix-vector operations between $\hat{X}$ and constants independent of the data, the calculation is fully differentiable with respect to the generator. Altogether, we can add the following loss term to the fine-tuning loss of $g_\theta$ to enforce $\phi$: $\mathcal{L}_\phi(g_\theta(z)) := \sum_i^N b_{\neg\phi}(g_\theta(z))_i$, using the notation $b_{\neg\phi}(g_\theta(z))$ for the binary mask calculated over the sample obtained from $g_\theta$.

Further, we extend the above relaxation to support logical implications, such as line 4 in Figure 3. We enforce implications $\phi \implies \psi$ over the $g_\theta$ by penalizing every generated row that violates the implication, *i.e.,* every row that satisfies $\zeta := \phi \wedge \neg\psi$. Notice that $\zeta$ can be understood as a row constraint expression, allowing for the techniques described above to calculate $b_\zeta(g_\theta(z))$ (note that we do not negate $\zeta$). Therefore, the resulting loss term to be added to the fine-tuning objective is:

$$\mathcal{L}_{\phi \implies \psi}(g_\theta(z)) := \sum_{i=1}^N b_\zeta(g_\theta(z))_i = \sum_{i=1}^N b_\phi(g_\theta(z))_i \odot b_{\neg\psi}(g_\theta(z))_i. \tag{2}$$

To *guarantee* that each sample respects the defined logical constraints, we use the same masking technique as during training to reject any generated sample that violates the constraint. In Section 5 we demonstrate that the described fine-tuning with the relaxed constraints is necessary to achieve high performance, with rejection sampling alone not being sufficient.

**Statistical Customization**    One may want to smoothen out undesired statistical differences between certain groups to limit bias, *e.g.,* encourage that the mean age measured over males and females agree (line 5 of Figure 3); or obfuscate sensitive statistical information, such as hiding the most prevalent disease in their dataset (recall the example in Section 1). To facilitate such statistical customizability we support the calculation of conditional statistical operations (expectation, variance, standard deviation, and entropy) composed into arithmetic ($+$, $-$, $*$, $/$) and logical ($\wedge$, $\vee$, $<$, $\leq$, $>$, $\geq$, $=$, $\neq$) expressions. The calculation of the corresponding loss term consists of two steps: (i) differentiably calculating the value of each involved statistical expression (involved), and (ii) as afterwards we are left with logical and arithmetical terms of reals, we can calculate the resulting loss term using t-norms and DL2 primitives (Fischer et al., 2019). Here (ii), we rely on prior work (Fischer et al., 2019), therefore we only elaborate on the more involved step (i) below.

Denote a conditional statistical operator as $OP[f(\mathcal{S})|\phi]$, where $f$ is a differentiable function over a subset of features $\mathcal{S}$, and $\phi$ is a row constraint condition. Incorporating such an expression is fundamentally challenging, as the conditioning is not differentiable. To address this issue, we select all rows of $\hat{X}$ where $\phi$ applies, using the differentiable technique for row constraints, described in an earlier paragraph. From the resulting subset of the sample $\hat{X}_\phi \subseteq \hat{X}$, we compute the normalized joint marginal of all features involved in $\mathcal{S}$, $\bar{\mu}(\mathcal{S}, \hat{X}_\phi)$, describing a probability distribution over $f(\mathcal{S})$, which enables the computation of the given statistical operation following its mathematical definition.

As such statistical specifications can be directly measured on any produced sample, they can be verified by the sampling entity if they are met to a desired degree or further regularization is needed using the soft-constraining procedure described above. Section 5 we demonstrate that the above procedure allows for effective statistical customization while preserving high synthetic data quality.

**Downstream Specifications**    As the synthetic data is expected to be deployed to train machine learning models, we need to support specifications involving them. For instance, consider synthetic data such that models trained on it exhibit lower bias, or that no models can be trained on the data to predict a certain protected column (lines 6 and 7 of Figure 3). Facilitating such specifications is challenging, as here we have to optimize not over measures of the data itself, but instead over the *effect* of the data on downstream classifiers. We achieve this by introducing a novel regularizer involving the differentiable training of downstream models. In each iteration of fine-tuning $g_\theta$, we train a differentiable surrogate classifier $h_\psi$ on the prediction task defined by the provided specification. Then, we "test" $h_\psi$ on the reference dataset $X_r$, and compute the statistic of interest $SI$ (*e.g.,* demographic parity distance $\pi_{\mathcal{D}_s}$ for bias w.r.t. the protected feature $\mathcal{D}_s$, or the cross entropy $\mathcal{L}_{CE}$ for predictive objectives). We then update $g_\theta$ influencing $SI$ in our desired direction. Denote the synthetic sample generated at the current iteration as $\hat{X}$, the features available to the surrogate model for prediction as $\hat{X}[\texttt{features}]$, and the target features as $\hat{X}[\texttt{target}]$. Then the loss term added to the fine-tuning objective can be defined as:

$$\mathcal{L}_{\text{DOWNSTREAM}}(g_\theta(z), X_r) \coloneqq s \cdot SI(h_{\psi^*}(X[\texttt{features}]), X[\texttt{target}]), \tag{3}$$

with

$$\psi^* \coloneqq \min_\psi \mathcal{L}_{CE}(h_\psi(\hat{X}[\texttt{features}]), \hat{X}[\texttt{target}]), \tag{4}$$

where $\mathcal{L}_{CE}$ denotes the cross-entropy loss, and $s \in \{-1, 1\}$ depending on whether we wish to maximize or minimize the computed statistic. Note that $\psi^*$ depends (differentiably) on $\theta$ through $\hat{X}$, and as such Equation (3) exhibits a differentiable dependency on $\theta$.

In Section 5 we demonstrate the effectiveness of the above method in encouraging desirable behavior from downstream models and even setting a new state-of-the-art in fair synthetic data generation.

## 5    EXPERIMENTAL EVALUATION

**Experimental Setup**    For realizing $g_\theta$ we use a fully connected neural network with residual connections. The regularization parameters are selected with on a hold-out validation dataset. Wherever possible, we report the mean and standard deviation of a given metric, measured over 5 retrainings and 5 samples from each model. For further details on the experimental setup please see Appendix A. We evaluate our method on four popular tabular datasets: Adult (Dua & Graff, 2017), German Credit (Dua & Graff, 2017), Compas (Angwin et al., 2016), and the Health Heritage Prize dataset from Kaggle (Kaggle, 2023). Due to the space constraint, most experiments included in the main paper are conducted on the Adult dataset, and repeated on *all* other datasets in Appendix B. For evaluating the quality of the produced synthetic data w.r.t., we measure the test accuracy of an XGBoost (Chen & Guestrin, 2016) model trained on the synthetic data and tested on the real test data. We resort to this evaluation metric to keep the presentation compact, while providing a comprehensive measure of the usefulness of the generated data. As XGBoost is state-of-the-art on tabular classification problems, it allows us to capture fine-grained deviations in data quality (Kotelnikov et al., 2022). We compare only to prior works with an available open source implementation.

**Downstream Specifications: Reducing Bias and Predictability**    We evaluate ProgSyn's performance on the task of generating a synthetic copy of the Adult dataset that is fair w.r.t the `sex` feature, both in the non-private and private (DP) setting, using the command shown in line 6 of Figure 3. We compare to two recent non-private (DECAF (van Breugel et al., 2021), and TabFairGAN (Rajabi & Garibay, 2022)), and one private (Prefair (Pujol et al., 2022)) fair synthetic data generation methods. The statistic of interest is a low demographic parity distance w.r.t. the `sex` feature of an XGBoost trained on the synthetic dataset and tested on the real testing dataset. In Table 1, we collect both our results in the non-private (top) and in the private (bottom, $\epsilon = 1$) settings. Notice that ProgSyn attains the highest accuracy and lowest demographic parity distance in both settings, achieving a new state-of-the-art in both private and non-private fair synthetic data generation. Most notably, while the

other methods were specifically developed for producing fair synthetic data, ProgSyn is general, with this being just one of the many specifications it supports.

Further, it can often be useful to data owners to ensure that malicious actors cannot learn to predict certain personal features from the released synthetic data. Using the `DOWNSTREAM` command shown in line 7 of Figure 3, we synthesize Adult, such that it cannot be used to train a classifier predicting the `sex` feature from other columns. As a result, we reduce the balanced accuracy of an XGBoost on the `sex` feature from $83.3\%$ to $50.2\%$, *i.e.,* to random guessing, while retaining $84.4\%$ accuracy on the original task.

Table 1: XGB accuracy [%] vs. demographic parity distance on the `sex` feature of various fair synthetic data generation algorithms compared to ProgSyn, both in a non-private (top) and private ($\epsilon = 1$) settings (bottom).

|  | XGB Acc. [%] | Dem. Parity `sex` |
|---|---|---|
| True Data | $85.4 \pm 0.0$ | $0.18 \pm 0.00$ |
| DECAF Dem. Parity | $66.8 \pm 7.0$ | $0.08 \pm 0.07$ |
| TabFairGAN | $79.8 \pm 0.5$ | $0.02 \pm 0.01$ |
| ProgSyn | $\mathbf{82.1 \pm 0.3}$ | $\mathbf{0.01 \pm 0.01}$ |
| Prefair Greedy ($\epsilon = 1$) | $80.2 \pm 0.4$ | $0.04 \pm 0.01$ |
| Prefair Optimal ($\epsilon = 1$) | $75.7 \pm 1.5$ | $0.03 \pm 0.02$ |
| ProgSyn ($\epsilon = 1$) | $\mathbf{80.9 \pm 0.3}$ | $\mathbf{0.01 \pm 0.01}$ |

**Statistical Properties**  Recall that ProgSyn allows direct manipulations of statistical properties of the generated datasets, using `STATISTICAL` specifications. We evaluate its effectiveness on this task with 3 statistical commands on Adult: S1: set the average age across the dataset to 30 instead of the original $\approx 37$; S2: set the average age of males and females equal (line 5 in Figure 3); and S3: set the correlation of `sex` and `salary` to zero, *i.e.,* $\frac{\mathbb{E}[\text{sex}\cdot\text{salary}]-\mathbb{E}[\text{sex}]\,\mathbb{E}[\text{salary}]}{\sqrt{\text{Var}(\text{sex})\,\text{Var}(\text{salary})}} = 0$, which is easily expressible in ProgSyn. Note here we do not compare to prior work, as no prior work allows for such statistical manipulations. On S1, we achieve a mean age of $30.2$ retaining $84.6\%$ accuracy, while on S2 ProgSyn reduces the average age gap from 2.3 years to $< 0.1$ maintaining $85.1\%$ accuracy. Most interestingly, on S3, we reduce the correlation between `sex` and `salary` from $-0.2$ to just $-0.01$, and retain an impressive $84.9\%$ accuracy. We provide more details in Appendix D.

**Logical Constraints**  We evaluate the performance of ProgSyn in enforcing logical constraints on the Adult dataset, using three implication (I1, I2, I3) and two row constraints (RC1, RC2). While RC2 and I2 correspond to lines 3 and 4 in Figure 3, we list the rest of the constraints in Appendix D. Note that the binary mask obtained for each constraint, as explained in Section 4.2, can easily be used for rejection sampling (RS) from ProgSyn. Therefore, in our comparison, we distinguish between ProgSyn with just RS and ProgSyn fine-tuned on the given constraint and rejection sampled (FT + RS). In the private setting, we compare our performance also to AIM (McKenna et al., 2022), where we encode the constraints in the graphical model as structural zeros (SZ). We summarize our results in Table 2, where in the first two rows we show the constraint satisfaction rates (CSR) on the original dataset, and on the evaluated synthetic datasets (*i.e.,* we compare the methods at $100\%$ CSR). Observe that while other methods also yield competitive results on constraints that are easy to enforce, *i.e.,* have high base satisfaction rate, as the constraint difficulty increases, fine-tuning becomes necessary, yielding superior results. Further, we tested ProgSyn in case *all* 5 constraints are applied at once, resulting in $84.0\%$ accuracy, demonstrating a strong performance in composability. These experiments show that ProgSyn is strongly effective in enforcing logical constraints.

**Stacking Specifications of Different Types**  In a significantly harder scenario, the user may wish to conduct several customizations of different type simultaneously. To evaluate ProgSyn in this case, we selected at least one command from each of the previously examined ones, and combined them in a single ProgSyn program. We picked the following commands for this experiment: (i) the command used to generate fair synthetic data w.r.t. `sex`; (ii) & (iii) S1 and S2 statistical manipulations, setting the average age to thirty, and equating the average ages of males and females; and (iv) & (v) two logical implication constraints (I3 and I2) from Table 2. In Table 3 we show the effect of applying these customizations increasingly one-after-another, with each row in the table standing for one additional active specification (marked in green). Observe that after sacrificing the expected $\approx 3.4\%$ accuracy for achieving low bias, ProgSyn maintains stable accuracy, while adhering to all remaining customizations. This result demonstrates the strong ability of ProgSyn to effectively incorporate diverse specifications simultaneously, with little cost to synthetic data quality.

Table 2: XGB accuracy [%] of synthetic data at 100% constraint satisfaction rate (CSR) on three implication constraints (I1 - I3) and two row constraints, applied separately, both in a non-private (top) and private ($\epsilon = 1$) setting (bottom). RS: rejection sampling, FT: fine-tuning, and SZ: structural zeros. ProgSyn + FT + RS is consistent across all settings, maintaining high data quality throughout.

| Constraint | I1 | I2 | I3 | RC1 | RC2 |
| Real data CSR | 93.6% | 100% | 60.4% | 32.4% | 40.5% |
|---|---|---|---|---|---|
| ■ TVAE | $82.1 \pm 0.5$ | $82.1 \pm 0.5$ | $82.1 \pm 0.5$ | $81.1 \pm 1.2$ | $81.3 \pm 0.6$ |
| ■ CTGAN | $83.4 \pm 0.3$ | $83.0 \pm 0.6$ | $83.4 \pm 0.3$ | $82.5 \pm 0.7$ | $82.5 \pm 0.8$ |
| ■ ProgSyn + RS | $\mathbf{85.1 \pm 0.1}$ | $\mathbf{85.1 \pm 0.1}$ | $\mathbf{85.1 \pm 0.2}$ | $82.9 \pm 0.8$ | $84.5 \pm 0.1$ |
| ■ ProgSyn + FT + RS | $\mathbf{85.1 \pm 0.1}$ | $85.0 \pm 0.18$ | $\mathbf{85.1 \pm 0.2}$ | $\mathbf{84.7 \pm 0.1}$ | $\mathbf{84.8 \pm 0.2}$ |
| ■ AIM + SZ ($\epsilon = 1$) | $\mathbf{84.2 \pm 0.2}$ | $\mathbf{84.1 \pm 0.3}$ | $83.7 \pm 0.3$ | $73.9 \pm 0.8$ | $67.6 \pm 1.4$ |
| ■ ProgSyn + RS ($\epsilon = 1$) | $83.7 \pm 0.2$ | $83.7 \pm 0.2$ | $83.7 \pm 0.2$ | $81.0 \pm 0.9$ | $\mathbf{83.5 \pm 0.2}$ |
| ■ ProgSyn + FT + RS ($\epsilon = 1$) | $83.8 \pm 0.2$ | $83.7 \pm 0.2$ | $\mathbf{83.9 \pm 0.1}$ | $\mathbf{83.1 \pm 0.2}$ | $83.4 \pm 0.2$ |

Table 3: ProgSyn's performance on 5 different specifications applied together, progressively adding more of them. In each row the active specifications are highlighted in green. The specifications are: the command used for fair data; statistical manipulations S1 and S2, setting the average age to 30, and equating the average ages of males and females; and two implications. ProgSyn demonstrates strong composability, adhering to all customizations while maintaining competitive accuracy.

| XGB Acc. [%] | Dem. Parity `sex` | $\Delta$ Avg. Age to 30 | $\Delta$ M-F Avg. Age | I3 Sat. [%] | I2 Sat. [%] |
|---|---|---|---|---|---|
| $85.1 \pm 0.16$ | $0.19 \pm 0.005$ | $37.3 \pm 0.05$ | $2.3 \pm 0.17$ | $59.3 \pm 0.85$ | $98.5 \pm 0.09$ |
| $81.7 \pm 0.25$ | $0.02 \pm 0.007$ | $37.3 \pm 0.05$ | $2.1 \pm 0.19$ | $57.6 \pm 0.78$ | $96.7 \pm 0.09$ |
| $82.5 \pm 0.76$ | $0.06 \pm 0.053$ | $30.2 \pm 0.04$ | $1.3 \pm 0.14$ | $57.0 \pm 0.84$ | $96.4 \pm 0.25$ |
| $82.0 \pm 0.50$ | $0.04 \pm 0.036$ | $30.2 \pm 0.03$ | $0.0 \pm 0.10$ | $56.9 \pm 1.11$ | $96.5 \pm 0.19$ |
| $81.3 \pm 0.34$ | $0.01 \pm 0.006$ | $30.2 \pm 0.04$ | $0.0 \pm 0.12$ | $100.0 \pm 0.00$ | $95.5 \pm 0.16$ |
| $81.6 \pm 0.29$ | $0.02 \pm 0.011$ | $30.2 \pm 0.04$ | $0.1 \pm 0.12$ | $100.0 \pm 0.00$ | $100.0 \pm 0.00$ |

**Experiments on Health Heritage, German Credit, and Compas**   We demonstrate the generalizability of ProgSyn by repeating our main experiments presented above on three further tabular datasets. For each dataset, we defined 3 implication, 2 row constraint, 2 statistical, and one downstream fairness command. We then evaluated ProgSyn under the same setup as on the Adult dataset, comparing to baseline methods. Our detailed results are included in Appendix B, where ProgSyn exhibits competitive performance across all examined datasets. Most notably, it often prevails as the best method in fair synthetic data generation, outperforming state-of-the-art specialized approaches, both in the non-private and DP settings. Further, we draw similar conclusions from the experiments on these datasets as on Adult; namely, (i) for harder to enforce logical constraints soft-constrained fine-tuning benefits performance; (ii) ProgSyn can effectively facilitate diverse customizations at the same time. For further details on the results of the experiments on the Health Heritage, German Credit, and Compas datasets, we refer the reader to Appendix B.

## 6   CONCLUSION

We presented ProgSyn, a new method for programmable synthetic data generation. The key idea was to pretrain a generative model, and then fine-tune it according to programmable specifications provided by the data owner. The fine-tuning is performed by converting the specifications into a differentiable loss using novel relaxations. ProgSyn is first to enable data owners to customize their synthetic data to their own use case by programmatically declaring logical and statistical specifications. We evaluated ProgSyn on a large number of practical specifications, most of them not supported by prior work, and obtained strong results across several datasets. Moreover, on tasks supported by prior work we either match or exceed their performance, *e.g.,* we set a new state-of-the-art in fair synthetic data generation. Our work shows that it is possible to generate high-quality customized synthetic data, thus opening doors for its wider adoption.

## 7 REPRODUCIBILITY STATEMENT

Our submission includes an anonymized code repository, with details for reproducing all our main results presented in the paper. Further, in all our experiments we have accounted for randomness by running each experiment on several random seeds. Finally, both in the main paper and in the Appendix, we give details on the experimental setup used, and how and which hyperparameters were selected.

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

APPENDIX

In Appendix A we give extended details on the experimental setup used to evaluate ProgSyn, including hyperparameters, and training details. In Appendix B we present our main results on the Health Heritage, Compas, and German datasets. Appendix C presents additional results in private and non-private unconstrained settings on all four datasets, compared to six baselines. In Appendix D, we list all ProgSyn commands used for our evaluation in the main paper, together with their corresponding hyperparameters and method of selecting these. In Appendix E, we give the technical details of the private training method for ProgSyn. In Appendix F we explain the differences between the base generative model used in ProgSyn and GEM (Liu et al., 2021). Finally, in Appendix G and Appendix H, we discuss the broader impact and limitations of ProgSyn.

## A  EXTENDED EXPERIMENTAL DETAILS

In this section we give extended details on the experimental setup used to obtain our presented results, and introduce the datasets used in the main body of the paper, the UCI Adult Census dataset (Dua & Graff, 2017), the Health Heritage Prize dataset from Kaggle (Kaggle, 2023), the Compas dataset (Angwin et al., 2016), and the German Credit dataset (Dua & Graff, 2017).

### A.1  SETUP AND TRAINING PARAMETERS

Here, we first give more details on the experimental setup used to obtain the results presented in the main body of the paper. Then, we also list all parameters and their choices relevant for training ProgSyn. Finally, we list the reproduced baselines and link to their source code.

**Experimental Setup**  In each of our experiments the base architecture of the ProgSyn generative model $g_\theta$ is formed by a four-layer fully connected neural network with residual connections, where the first hidden layer contains 100 neurons, and the rest of the layers 200. The input dimension of the network, *i.e.,* the dimension of the sampled Gaussian noise $z$, is 100. In the non-private setting, we pre-train the generator for 2 000 epochs on a marginal workload containing all three-way feature marginals that involve the label. Then, we fine-tune on each constraint for a varying number of epochs using the original dataset as a reference (we give more training details for each constraint in Appendix D). In the private setting, we pre-train the generator on a marginal workload containing all three-way marginals in the dataset using our modified AIM algorithm presented in Appendix E. Then, we fine-tune on the constraints (we give more details for each constraint in Appendix D), using a sample from the model before fine-tuning for reference. We pre-train for each dataset and privacy scenario a generative model on random seed 42 and fine-tune it over 5 retries for each constraint. Then, from each of these models, we sample 5 datasets to measure the performance. Finally, we report the mean and the standard deviation of the resulting 25 measurements whenever possible. Note that this estimate incorporates the randomness in the fine-tuning phase, and the sampling noise.

**Hyperparameters**  For pre-training the non-private model, we use batch size 15 000 (*i.e.,* the generated dataset we measure the marginals of has 15 000 rows), and train the model for 2 000 epochs. For the private model, we use a batch size of 1 000, and train the generative model at each step of the private outer selection loop for 1 000 epochs. In both cases, we use the Adam optimizer, with the default parameters, in combination with the CosineAnnealing learning rate scheduler. Additionally, for non-private pre-training, we update on every group of 16 marginals, where one epoch is completed once we have updated on every marginal of all three-way marginals containing the label. For measuring the utility of the dataset using the XGB accuracy metric, we use an XGBoost classifier with the default hyperparameters, as included in the XGBoost Python library[1].

**Resources Used**  For running the experiments we had $7\times$ NVIDIA GeForce RTX 2080 Ti GPUs, $4\times$ NVIDIA TITAN RTX, $2\times$ NVIDIA GeForce GTX 1080 Ti, and $2\times$ NVIDIA A100 SXM 40GB Tensor Core GPUs available, where the A100 cards were used only for the experiments on Health Heritage.

---

[1]XGBoost library: https://xgboost.readthedocs.io/en/stable/python/python_api.html

**Reproducing Baselines**  Here, we will list the works we compared against, including a link to the repositories we have downloaded their code from. In our paper (including this appendix) we reproduced the following works for comparison:

- TVAE (Xu et al., 2019b), code from the Synthetic Data Vault Patki et al. (2016): `https://github.com/sdv-dev/SDV`,

- CTGAN (Xu et al., 2019b), code from the Synthetic Data Vault Patki et al. (2016): `https://github.com/sdv-dev/SDV`,

- GReaT (Borisov et al., 2023), code: `https://github.com/kathrinse/be_great`,

- AIM (McKenna et al., 2022), code: `https://github.com/ryan112358/private-pgm`,

- MST (McKenna et al., 2021), code: `https://github.com/ryan112358/private-pgm`,

- GEM (Liu et al., 2021), code: `https://github.com/terranceliu/iterative-dp`,

- Prefair (Pujol et al., 2022), code: `https://github.com/David-Pujol/Prefair`,

- DECAF (van Breugel et al., 2021), code from a reproduction study (Wang et al., 2022) (downloadable from the supplementary materials on OpenReview: `https://openreview.net/forum?id=SVx46hzmhRK`),

- TabFairGAN (Rajabi & Garibay, 2022), code: `https://github.com/amirarsalan90/TabFairGAN`.

## A.2 DATASETS

In this subsubsection we briefly describe the technical details of each dataset used in this paper.

**Adult**  The UCI Adult Census dataset (Dua & Graff, 2017) contains US-census data of $45\,222$ individuals (excluding incomplete rows), split into training and test sets of size $30\,162$ and $15\,060$, respectively. After removing the duplicate feature of `education_num`, the dataset contains 14 features (5 continuous and 9 discrete). We discretize each continuous feature uniformly in 32 bins and one-hot encode the data, resulting in 261 dimensions for each row. The original task of Adult is to predict the binary label of `salary`, which is 0 if the given individual earns $\geq \$50K$ per year, and 1 otherwise. The labels are imbalanced, with around $75\%$ of the labels being 1. This also means that any classifier that assigns the label 1 to every instance will have an accuracy of around $75\%$.

**Health Heritage**  The Health Heritage Prize dataset from Kaggle (Kaggle, 2023) contains health-related data of patients admitted to the hospital, collected in a table. The dataset is widely used in algorithmic fairness research in the machine learning community. The preprocessing details of the dataset are included in the accompanying code repository. The constructed task, in this case, is to classify each patient if they are likely to be admitted to emergency care in the near future or not, *i.e.,* if they have a `maxCharlsonIndex` of $> 0$ or $= 0$, respectively. The dataset contains $218\,415$ rows, where we randomly split to create a training dataset of $174\,732$ rows and a test set of $43\,683$ rows. There are 18 columns in the dataset, with 7 discrete and 11 continuous columns, where, again, we uniformly discretize the continuous columns into 32 bins. The dataset is imbalanced, with $\approx 64\%$ of the labels being $= 0$, therefore, a majority classifier achieves an accuracy of around $64\%$.

**Compas**  The Compas dataset (Angwin et al., 2016) contains personal attributes and criminal record related data of $6\,172$ individuals. The dataset is widely used in the fairness literature. To preprocess the dataset, we follow the same technique as Balunovic et al. (2022). Finally, we split the dataset into $4\,937$ training data points, and $1\,235$ testing data points. The dataset contains 9 columns, of which 5 are discrete and 4 are continuous. We discretize the continuous features into 32 equal-width bins. The dataset is relatively balanced, with around $55\%$ of the data points having label 1, therefore a classifier always predicting 1 only achieves an accuracy of around $55\%$.

Table 4: XGB accuracy [%] vs. demographic parity distance on the `AgeAtFirstClaim` feature of various fair synthetic data generation algorithms compared to ProgSyn on the **Health Heritage dataset**, both in a `non-private` (top) and `private` ($\epsilon = 1$) settings (bottom).

|  | XGB Acc. [%] | Dem. Parity `AgeAtFirstClaim` |
|---|---|---|
| True Data | $81.0 \pm 0.00$ | $0.51 \pm 0.000$ |
| ■ TabFairGAN | $78.7 \pm 0.45$ | $0.40 \pm 0.016$ |
| ■ ProgSyn | $70.9 \pm 0.67$ | $0.14 \pm 0.023$ |
| ■ Prefair Greedy ($\epsilon = 1$) | $73.5 \pm 0.11$ | $0.35 \pm 0.004$ |
| ■ Prefair Optimal* ($\epsilon = 1$) | - | - |
| ■ ProgSyn ($\epsilon = 1$) | $73.9 \pm 0.17$ | $0.228 \pm 0.006$ |

Table 5: XGB accuracy [%] vs. demographic parity distance on the `race` feature of various fair synthetic data generation algorithms compared to ProgSyn on the **Compas dataset**, both in a `non-private` (top) and `private` ($\epsilon = 1$) settings (bottom).

|  | XGB Acc. [%] | Dem. Parity `race` |
|---|---|---|
| True Data | $63.4 \pm 0.00$ | $0.13 \pm 0.000$ |
| ■ TabFairGAN | $62.3 \pm 2.36$ | $0.19 \pm 0.057$ |
| ■ ProgSyn | $62.1 \pm 1.28$ | $0.05 \pm 0.031$ |
| ■ Prefair Greedy ($\epsilon = 1$) | $59.4 \pm 1.53$ | $0.15 \pm 0.058$ |
| ■ Prefair Optimal ($\epsilon = 1$) | $58.0 \pm 3.31$ | $0.13 \pm 0.055$ |
| ■ ProgSyn ($\epsilon = 1$) | $60.5 \pm 0.58$ | $0.04 \pm 0.032$ |

**German Credit**  The German Credit dataset (Dua & Graff, 2017) contains personal data of $1\,000$ individuals, where the task is to classify each person in good or bad credit risk. We randomly split the dataset into 800 training data points and 200 test data points. The dataset consists of 20 columns, of which 14 are categorical, and the rest are continuous, which we discretize into 32 equal-width bins. The dataset is imbalanced, with approximately 70% of the labels being 0, therefore, a classifier predicting only 0 achieves $\approx 70\%$ accuracy.

## B    MAIN RESULTS ON GERMAN CREDIT, COMPAS, AND HEALTH HERITAGE

### B.1    FAIRNESS

In Tables 4 to 5 we present our results on fair synthetic data generation on the Health Heritage, Compas, and German datasets, respectively. Notice that ProgSyn exhibits a consistently strong performance, often clearly providing the best accuracy-fairness trade-off. Note also that in some cases Prefair Optimal (Pujol et al., 2022) did not converge even after more than a week of running. Also, in the DP case, on German ProgSyn due to the low prevalence of the protected class, the DP noise eliminated that class from the modeled distribution, and as such, no fairness measurements were possible. This is because the German dataset has very few samples (800) therefore it is possible that DP leads to the complete elimination of certain features. Note that for these experiments we binarize the `AgeAtFirstClaim` column of the Health Heritage dataset with patients above and below sixty, and we also binarize the `race` column of the Compas dataset by only keeping the `Caucasian` and `African-American` features. Note that here we follow the example of fair representation learning literature, *e.g.,* Balunovic et al. (2022).

### B.2    LOGICAL CONSTRAINTS

In Tables 7 to 9 we present our results on the Health Heritage, Compas, and German datasets in enforcing logical constraints. Notice that the observations that can be drawn from these tables match those made in the main paper; namely, ProgSyn outperforms the methods from the Synthetic Data Vault (Patki et al., 2016), and fine-tuning helps in enforcing hard logical constraints.

Table 6: XGB accuracy [%] vs. demographic parity distance on the `foreign worker` feature of various fair synthetic data generation algorithms compared to ProgSyn on the **German Credit dataset**, both in a non-private (top) and private ($\epsilon = 1$) settings (bottom).

|  | XGB Acc. [%] | Dem. Parity `foreign worker` |
|---|---|---|
| True Data | $74.0 \pm 0.00$ | $0.28 \pm 0.000$ |
| ■ TabFairGAN | $64.0 \pm 4.57$ | $0.09 \pm 0.064$ |
| ■ ProgSyn | $73.6 \pm 1.43$ | $0.10 \pm 0.091$ |
| ■ Prefair Greedy ($\epsilon = 1$) | $65.0 \pm 5.32$ | $0.12 \pm 0.086$ |
| ■ Prefair Optimal* ($\epsilon = 1$) | $62.5 \pm 2.07$ | $0.22 \pm 0.125$ |
| ■ ProgSyn ($\epsilon = 1$) | - | - |

Table 7: XGB accuracy [%] of synthetic data at $100\%$ constraint satisfaction rate (CSR) on three implication constraints (I1 - I3) and two row constraints, applied separately, both in a non-private (top) and private ($\epsilon = 1$) setting (bottom) on the **Health Heritage dataset**. RS: rejection sampling, and FT: fine-tuning. ProgSyn + FT + RS is consistent across all settings, maintaining high data quality throughout.

| Constraint Real data CSR | I1 18.3% | I2 79.3% | I3 5.4% | RC1 44.8% | RC2 1.8% |
|---|---|---|---|---|---|
| ■ TVAE | $78.2 \pm 0.22$ | $78.2 \pm 0.23$ | $77.8 \pm 0.40$ | $78.1 \pm 0.20$ | $68.8 \pm 6.70$ |
| ■ CTGAN | $78.4 \pm 0.22$ | $78.8 \pm 0.58$ | $78.5 \pm 0.41$ | $78.7 \pm 0.51$ | $75.9 \pm 2.62$ |
| ■ ProgSyn + RS | $80.0 \pm 0.08$ | $\mathbf{80.1 \pm 0.08}$ | $\mathbf{79.9 \pm 0.09}$ | $79.9 \pm 0.07$ | $79.2 \pm 0.11$ |
| ■ ProgSyn + FT + RS | $\mathbf{80.1 \pm 0.09}$ | $80.0 \pm 0.11$ | $79.7 \pm 0.12$ | $\mathbf{80.0 \pm 0.08}$ | $\mathbf{79.6 \pm 0.13}$ |
| ■ ProgSyn + RS ($\epsilon = 1$) | $\mathbf{77.9 \pm 0.09}$ | $77.9 \pm 0.13$ | $\mathbf{77.8 \pm 0.10}$ | $\mathbf{77.8 \pm 0.09}$ | $\mathbf{77.8 \pm 0.12}$ |
| ■ ProgSyn + FT + RS ($\epsilon = 1$) | $77.7 \pm 0.12$ | $\mathbf{78.1 \pm 0.11}$ | $77.6 \pm 0.13$ | $77.4 \pm 0.10$ | $77.7 \pm 0.09$ |

Table 8: XGB accuracy [%] of synthetic data at $100\%$ constraint satisfaction rate (CSR) on three implication constraints (I1 - I3) and two row constraints, applied separately, both in a non-private (top) and private ($\epsilon = 1$) setting (bottom) on the **Compas dataset**. RS: rejection sampling, and FT: fine-tuning. ProgSyn + FT + RS is consistent across all settings, maintaining high data quality throughout.

| Constraint Real data CSR | I1 60.4% | I2 87.0% | I3 26.2% | RC1 35.2% | RC2 51.8% |
|---|---|---|---|---|---|
| ■ TVAE | $66.2 \pm 1.03$ | $66.2 \pm 1.02$ | $66.2 \pm 1.13$ | $64.9 \pm 0.97$ | $63.9 \pm 0.99$ |
| ■ CTGAN | $60.4 \pm 2.30$ | $60.5 \pm 3.38$ | $59.9 \pm 3.64$ | $60.4 \pm 2.67$ | $59.1 \pm 2.30$ |
| ■ ProgSyn + RS | $\mathbf{65.2 \pm 0.89}$ | $\mathbf{65.2 \pm 0.83}$ | $\mathbf{64.8 \pm 1.01}$ | $64.2 \pm 1.38$ | $62.0 \pm 0.66$ |
| ■ ProgSyn + FT + RS | $64.1 \pm 0.82$ | $64.8 \pm 0.97$ | $61.1 \pm 0.87$ | $\mathbf{64.6 \pm 1.35}$ | $\mathbf{62.3 \pm 0.76}$ |
| ■ ProgSyn + RS ($\epsilon = 1$) | $\mathbf{62.7 \pm 0.93}$ | $\mathbf{62.7 \pm 0.95}$ | $\mathbf{62.4 \pm 1.15}$ | $\mathbf{62.6 \pm 0.82}$ | $\mathbf{60.3 \pm 0.73}$ |
| ■ ProgSyn + FT + RS ($\epsilon = 1$) | $61.9 \pm 0.86$ | $62.5 \pm 0.63$ | $58.4 \pm 1.51$ | $62.1 \pm 0.70$ | $59.1 \pm 0.77$ |

Table 9: XGB accuracy [%] of synthetic data at $100\%$ constraint satisfaction rate (CSR) on three implication constraints (I1 - I3) and two row constraints, applied separately, both in a non-private (top) and private ($\epsilon = 1$) setting (bottom) on the **German dataset**. RS: rejection sampling, and FT: fine-tuning. ProgSyn + FT + RS is consistent across all settings, maintaining high data quality throughout.

| Constraint | I1 | I2 | I3 | RC1 | RC2 |
|---|---|---|---|---|---|
| Real data CSR | 83.9% | 61.7% | 5.5% | 40.9% | 29.1% |
| ■ TVAE | $72.0 \pm 1.73$ | $71.3 \pm 1.80$ | $72.2 \pm 1.98$ | $72.1 \pm 1.66$ | $70.5 \pm 0.00$ |
| ■ CTGAN | $63.4 \pm 4.86$ | $63.8 \pm 3.24$ | $64.4 \pm 4.01$ | $64.0 \pm 4.28$ | $63.5 \pm 5.30$ |
| ■ ProgSyn + RS | $\mathbf{72.4 \pm 2.82}$ | $\mathbf{73.3 \pm 1.95}$ | $\mathbf{73.6 \pm 2.80}$ | $72.8 \pm 2.24$ | $71.2 \pm 1.80$ |
| ■ ProgSyn + FT + RS | $70.2 \pm 2.51$ | $73.0 \pm 2.13$ | $72.6 \pm 2.25$ | $\mathbf{73.7 \pm 2.29}$ | $\mathbf{72.6 \pm 2.36}$ |
| ■ ProgSyn + RS ($\epsilon = 1$) | $\mathbf{65.1 \pm 3.16}$ | $\mathbf{65.4 \pm 2.58}$ | $64.6 \pm 3.24$ | $\mathbf{68.7 \pm 2.00}$ | $60.6 \pm 3.94$ |
| ■ ProgSyn + FT + RS ($\epsilon = 1$) | $63.0 \pm 4.14$ | $65.1 \pm 2.89$ | $\mathbf{66.4 \pm 2.54}$ | $59.7 \pm 3.66$ | $\mathbf{64.8 \pm 2.96}$ |

Table 10: ProgSyn's performance on 5 different specifications applied together, progressively adding more of them on the **Health Heritage dataset**. In each row the active specifications are highlighted in green. The specifications are: the command used for fair data; statistical manipulations S1 and S2; and two implications. ProgSyn demonstrates strong composability, adhering to customizations while maintaining competitive accuracy.

| XGB Acc. [%] | Dem. Parity | S1 | S2 | I3 Sat. [%] | I2 Sat. [%] |
|---|---|---|---|---|---|
| $79.7 \pm 0.09$ | $0.52 \pm 0.005$ | $0.2 \pm 0.00$ | $5.7 \pm 0.01$ | $5.6 \pm 0.09$ | $77.5 \pm 0.79$ |
| $76.7 \pm 0.13$ | $0.33 \pm 0.005$ | $0.2 \pm 0.00$ | $5.7 \pm 0.02$ | $5.4 \pm 0.11$ | $81.4 \pm 2.35$ |
| $77.0 \pm 0.19$ | $0.34 \pm 0.008$ | $0.0 \pm 0.00$ | $5.7 \pm 0.02$ | $5.4 \pm 0.11$ | $81.8 \pm 2.28$ |
| $76.3 \pm 0.24$ | $0.32 \pm 0.014$ | $0.0 \pm 0.00$ | $0.0 \pm 0.00$ | $5.1 \pm 0.09$ | $79.8 \pm 2.12$ |
| $74.8 \pm 0.31$ | $0.24 \pm 0.011$ | $0.6 \pm 0.03$ | $0.0 \pm 0.00$ | $100.0 \pm 0.00$ | $95.9 \pm 0.63$ |
| $76.0 \pm 0.43$ | $0.33 \pm 0.011$ | $0.6 \pm 0.01$ | $0.0 \pm 0.00$ | $100.0 \pm 0.00$ | $100.0 \pm 0.00$ |

### B.3 STACKING SPECIFICATIONS

In Tables 5, 10 and 11 we show our results in chaining specifications on the Helath Heritage, Compas, and German datasets, respectively. Aligned with the conclusions drawn in the main part of this paper, ProgSyn proves to be an effective method in being able to deal with several specification simultaneously.

Table 11: ProgSyn's performance on 5 different specifications applied together, progressively adding more of them on the **Compas dataset**. In each row the active specifications are highlighted in green. The specifications are: the command used for fair data; statistical manipulations S1 and S2; and two implications. ProgSyn demonstrates strong composability, adhering to all customizations. Note that once I3 is introduced, a constraint with which the method seems to struggle already, the accuracy decreases by an amount expected after Table 8.

| XGB Acc. [%] | Dem. Parity | Mean Age to 40 (S1) | Cov(sex, $y$) (S2) | I3 Sat. [%] | I2 Sat. [%] |
|---|---|---|---|---|---|
| $63.7 \pm 1.05$ | $0.20 \pm 0.042$ | $33.6 \pm 0.17$ | $0.1 \pm 0.01$ | $26.2 \pm 1.02$ | $87.1 \pm 0.90$ |
| $61.6 \pm 1.09$ | $0.05 \pm 0.021$ | $33.5 \pm 0.15$ | $0.1 \pm 0.02$ | $27.3 \pm 0.79$ | $87.7 \pm 0.76$ |
| $61.9 \pm 0.94$ | $0.06 \pm 0.036$ | $39.6 \pm 0.19$ | $0.1 \pm 0.01$ | $26.4 \pm 0.69$ | $87.3 \pm 1.01$ |
| $61.4 \pm 1.04$ | $0.05 \pm 0.040$ | $39.8 \pm 0.21$ | $0.0 \pm 0.02$ | $26.3 \pm 0.86$ | $87.7 \pm 0.62$ |
| $54.3 \pm 1.89$ | $0.07 \pm 0.042$ | $39.1 \pm 0.18$ | $0.0 \pm 0.02$ | $100.0 \pm 0.00$ | $100.0 \pm 0.00$ |
| $53.9 \pm 1.57$ | $0.05 \pm 0.035$ | $39.1 \pm 0.20$ | $0.0 \pm 0.02$ | $100.0 \pm 0.00$ | $100.0 \pm 0.00$ |

Table 12: ProgSyn's performance on 5 different specifications applied together, progressively adding more of them on the **German dataset**. In each row the active specifications are highlighted in green. The specifications are: the command used for fair data; statistical manipulations S1 and S2; and two implications. ProgSyn demonstrates strong composability, adhering to all customizations while maintaining competitive accuracy.

| XGB Acc. [%] | Dem. Parity | Mean Age to 40 (S1) | Cov(prot.,$y$) (S2) | I3 Sat. [%] | I2 Sat. [%] |
|---|---|---|---|---|---|
| $73.7 \pm 2.86$ | $0.14 \pm 0.093$ | $34.7 \pm 0.31$ | $-0.1 \pm 0.04$ | $7.0 \pm 2.41$ | $62.1 \pm 3.53$ |
| $72.2 \pm 2.03$ | $0.09 \pm 0.064$ | $34.5 \pm 0.47$ | $-0.1 \pm 0.04$ | $6.8 \pm 2.24$ | $60.9 \pm 2.21$ |
| $72.9 \pm 3.06$ | $0.11 \pm 0.071$ | $40.0 \pm 0.40$ | $-0.1 \pm 0.05$ | $6.6 \pm 2.72$ | $62.8 \pm 2.73$ |
| $73.0 \pm 2.35$ | $0.09 \pm 0.050$ | $40.0 \pm 0.43$ | $0.0 \pm 0.04$ | $6.4 \pm 2.14$ | $61.6 \pm 3.11$ |
| $71.5 \pm 2.48$ | $0.10 \pm 0.097$ | $40.0 \pm 0.25$ | $-0.0 \pm 0.02$ | $100.0 \pm 0.00$ | $62.3 \pm 2.32$ |
| $71.4 \pm 2.85$ | $0.10 \pm 0.097$ | $40.7 \pm 0.31$ | $0.0 \pm 0.03$ | $100.0 \pm 0.00$ | $100.0 \pm 0.00$ |

Table 13: TV distance on the training marginals, and downstream XGB accuracy, comparing ProgSyn with baseline non-private generative models and private ($\epsilon = 1.0$) generative models on the **Adult dataset**. The true data leads to an XGB accuracy of $86.7\%$, and to $85.4\%$ when discretized. Per metric, we highlight the best model in **bold** and underline the second best.

| Non-Private | ProgSyn | TVAE | CTGAN | GReaT |
|---|---|---|---|---|
| TV distance [$\cdot 10^{-5}$] | $\mathbf{4.1 \pm 0.09}$ | $28.6 \pm 4.02$ | $34.2 \pm 2.43$ | $\underline{26.8 \pm 0.21}$ |
| XGB acc. [%] | $\underline{85.2 \pm 0.12}$ | $82.0 \pm 0.43$ | $83.3 \pm 0.32$ | $\mathbf{85.7 \pm 0.13}$ |

| Private ($\epsilon = 1$) | ProgSyn | MST | GEM | AIM |
|---|---|---|---|---|
| TV distance [$\cdot 10^{-5}$] | $\underline{8.8 \pm 0.19}$ | $13.9 \pm 0.22$ | $34.9 \pm 0.14$ | $\mathbf{7.1 \pm 0.14}$ |
| XGB acc. [%] | $\underline{83.5 \pm 0.26}$ | $79.7 \pm 0.61$ | $79.3 \pm 0.90$ | $\mathbf{84.1 \pm 0.33}$ |

## C  UNCONSTRAINED NON-PRIVATE AND PRIVATE GENERATION

In this subsection, we present our results in unconstrained non-private and private generation on both the Adult, Health Heritage, Compas, and German Credit datasets. For evaluation, we use two metrics: (i) the total variation distance between the training marginals and the marginals of the synthetic dataset, and (ii) the downstream XGB accuracy metric as used in the main body of the paper. The goal of these experiments is to understand the performance of the generative model underlying ProgSyn, note, however, that this generative model is not our main contribution. We believe that ProgSyn will greatly benefit from improvements to its generative backbone by future work.

**Non-Private Generation**   To understand the raw performance of $g_\theta$ we trained it on the Adult, Health Heritage, Compas, and German Credit datasets w.r.t. all three-way marginals that include the original task label. Then, we evaluated the synthetic data generated by this model and compared it to three state-of-the-art tabular synthetic data generators, TVAE (Xu et al., 2019b), CTGAN (Xu et al., 2019b), and GReaT (Borisov et al., 2023). Note that these models were designed with the sole purpose of generating non-private synthetic data as close to the real data as possible in performance. As such, they constitute a much more restricted set of models, that do not directly support DP training nor customizations. In the top halves of Tables 13 to 16 we collect our results comparing the performance of ProgSyn to the above-mentioned baselines. Note that on the Health Heritage and German Credit datasets, we do not report any results for GReaT as it did not generate even a single sample after 4+ hours of sampling that was accepted by the sampling filter in GReaT. Looking at the non-private results, we can observe that ProgSyn achieved an around $7\times$ reduction in TV distance on the target marginals compared to the next best non-private method GReaT on Adult, and more than $9\times$ on Health Heritage compared to CTGAN. However, it is important to note that in contrast to ProgSyn the other models do not directly optimize on the marginals. On XGB accuracy, ProgSyn ranks as a competitive second-best method behind GReaT on Adult, exhibiting a comfortable margin to TVAE and CTGAN; while on Health Heritage the comparison to GReaT was not possible, the margin to the other methods is still significant. On Compas, somewhat surprisingly, TVAE ranks as the best method, with GReaT and ProgSyn close in performance, while CTGAN is significantly

Table 14: TV distance on the training marginals, and downstream XGB accuracy, comparing ProgSyn with baseline non-private generative models and private ($\epsilon = 1.0$) generative models on the **Health Heritage dataset**. The true data leads to an XGB accuracy of 81.3%, and to 81.1% when discretized. Per metric, we highlight the best model in **bold** and underline the second best.

| Non-Private | ProgSyn | TVAE | CTGAN | GReaT |
|---|---|---|---|---|
| TV distance [$\cdot 10^{-5}$] | **1.04 ± 0.01** | 12.3 ± 1.44 | 9.6 ± 0.42 | – |
| XGB acc. [%] | **80.1 ± 0.07** | 78.2 ± 0.22 | 78.3 ± 0.65 | – |
| Private ($\epsilon = 1$) | ProgSyn | MST | GEM | AIM |
| TV distance [$\cdot 10^{-5}$] | 3.0 ± 0.05 | 4.3 ± 0.02 | 5.5 ± 0.08 | **1.5 ± 0.03** |
| XGB acc. [%] | 77.9 ± 0.13 | 74.2 ± 0.10 | 76.5 ± 0.21 | **80.2 ± 0.09** |

Table 15: TV distance on the training marginals, and downstream XGB accuracy, comparing ProgSyn with baseline non-private generative models and private ($\epsilon = 1.0$) generative models on the **Compas dataset**. The true data leads to an XGB accuracy of 69.9%, and to 67.0% when discretized. Per metric, we highlight the best model in **bold** and underline the second best.

| Non-Private | ProgSyn | TVAE | CTGAN | GReaT |
|---|---|---|---|---|
| TV distance [$\cdot 10^{-4}$] | **2.12 ± 0.17** | 20.2 ± 1.95 | 18.4 ± 2.9 | 7.74 ± 0.35 |
| XGB acc. [%] | 65.6 ± 0.78 | **66.7 ± 1.04** | 60.8 ± 2.28 | 65.7 ± 1.14 |
| Private ($\epsilon = 1$) | ProgSyn | MST | GEM | AIM |
| TV distance [$\cdot 10^{-4}$] | 5.87 ± 0.77 | 8.88 ± 1.17 | 29.2 ± 2.8 | **3.64 ± 0.15** |
| XGB acc. [%] | 61.9 ± 2.13 | 62.6 ± 1.30 | 56.0 ± 1.77 | **64.0 ± 0.81** |

worse. Meanwhile on the German Credit dataset, ProgSyn ranks as the best method. These results argue that the ProgSyn backbone is a strong generative model for tabular data.

**Private Generation** We compare the DP trained ProgSyn backbone to three state-of-the-art DP methods, MST (McKenna et al., 2021), AIM (McKenna et al., 2022), and GEM (Liu et al., 2021) on the Adult, Health Heritage, Compas, and German datasets at a privacy level of $\epsilon = 1$. Note that as all three of these baseline models require the same kind of discretization as ProgSyn, the comparison is fair without further adjustments. We show our results in the bottom half of Tables 13 and 14. Observe that ProgSyn often ranks as a strong second-best method behind AIM, more often than not exhibiting a fair margin to the other methods. Most notably, on the German Credit dataset, it ranks as the best method based on the XGB accuracy. This is remarkable, as AIM is, to the best of our knowledge, the strongest currently available DP synthetic data generation model, but is far less versatile than ProgSyn, which supports non-private training and a large set of constraints. Altogether, this experiment demonstrates that ProgSyn is a strong base generative model for fine-tuning on constraints, even in the private setting.

Table 16: TV distance on the training marginals, and downstream XGB accuracy, comparing ProgSyn with baseline non-private generative models and private ($\epsilon = 1.0$) generative models on the **German dataset**. The true data leads to an XGB accuracy of 78.0%, and to 74.0% when discretized. Per metric, we highlight the best model in **bold** and underline the second best.

| Non-Private | ProgSyn | TVAE | CTGAN | GReaT |
|---|---|---|---|---|
| TV distance [$\cdot 10^{-4}$] | **5.44 ± 0.19** | 37.1 ± 1.00 | 13.6 ± 0.66 | – |
| XGB acc. [%] | **73.6 ± 1.64** | 69.7 ± 1.88 | 64.0 ± 3.65 | – |
| Private ($\epsilon = 1$) | ProgSyn | MST | GEM | AIM |
| TV distance [$\cdot 10^{-3}$] | 2.9 ± 0.21 | **1.7 ± 0.09** | 6.43 ± 0.18 | 1.78 ± 0.1 |
| XGB acc. [%] | **67.0 ± 2.88** | 63.3 ± 2.78 | 62.3 ± 16.40 | 64.1 ± 3.66 |

## D  CONSTRAINT EXPERIMENTS

In this section we first explain how one can choose the weights for the constraints without violating the condition of train-test separation. Then, we list all commands used for the experiments in the

Table 17: Commands and hyperparameters used in the experiment:
**Downstream Constraints: Eliminating Bias and Predictability**

| Command | Weights |
|---|---|
| **Reducing bias, non-private:**
```SYNTHESIZE: Adult;```
``` MINIMIZE: FAIRNESS:```
``` DEMOGRAPHIC_PARITY(protected=sex, target=salary,```
``` lr=0.1, n_epochs=15, batch_size=256);```
```END;``` | 0.0009 |
| **Reducing bias, private ($\epsilon = 1$):**
```SYNTHESIZE: Adult;```
``` ENSURE: DIFFERENTIAL PRIVACY:```
``` EPSILON=1.0, DELTA=1e-9;```
``` MINIMIZE: FAIRNESS:```
``` DEMOGRAPHIC_PARITY(protected=sex, target=salary,```
``` lr=0.1, n_epochs=15, batch_size=256);```
```END;``` | 0.0007 |
| **Predictability, non-private:**
```SYNTHESIZE: Adult;```
``` MINIMIZE: UTILITY:```
``` DOWNSTREAM_ACCURACY(features=all, target=salary);```
```END;``` | 0.00133 |

main paper, with their corresponding hyperparameters (constraint weight and number of fine-tuning epochs).

### D.1 CHOOSING THE CONSTRAINT WEIGHTS AND OTHER HYPERPARAMETERS

For choosing the constraint weights $\{\lambda_i\}_{i=1}^n$, we implemented a $k$-fold cross-validation scheme splitting over the reference dataset of the fine-tuning objective. We fine-tune for each $k$ splits and each weight that is to be evaluated. The results are reported for each weight combination and their corresponding diagnostic metrics on the data utility and the constraint satisfaction degree. The user can use this diagnostic data to gauge the weights they want to set in their constraint program for their final fine-tuning phase. To choose the weights we used for the results presented in the main body of the paper, in order to save time and compute, we did not run the full $k$-fold cross-validation, but validated only on the first split at $k = 5$, and chose the best performing parameter from this data.

### D.2 COMMANDS USED

In this subsection we list for each paragraph from the experimental section in the main body the corresponding commands and hyperparameters. Note that the syntax in the listed commands slightly differs from the syntax presented in the main paper. The reason for this is that the commands included here serve the purpose of reproduction, and therefore follow the syntax of the code repository version submitted in the supplementary materials. The code syntax in the paper uses a more intuitive syntax which is being adapted in the codebase in the current refactoring. For all constraints, we use batch size 15 000.

- **Downstream Constraints: Eliminating Bias and Predictability**: Table 17.
- **Statistical Properties**: Table 18.
- **Logical Constraints**: non-private: Table 19, private: Table 20.
- **Stacking Constraints of Different Types**: Table 21.

### E DIFFERENTIALLY PRIVATE TRAINING OF PROGSYN

In the case of DP training, we adapt the iterative and privacy budget adaptive DP training algorithm presented in AIM (McKenna et al., 2022). In brief, given a privacy budget $\epsilon$ and a workload (set of

Table 18: Commands and hyperparameters used in the experiment:
**Statistical Properties**

| Command | Weights |
|---|---|
| **Set the average age to 30 (S1):**
`SYNTHESIZE: Adult;`
`  ENFORCE: STATISTICAL:`
`    E[age] == 30;`
`END;` | 0.000025 |
| **Set the average age of males and females equal (S2):**
`SYNTHESIZE: Adult;`
`  ENFORCE: STATISTICAL:`
`    E[age\|sex==Male] == E[age\|sex==Female];`
`END;` | 0.0000125 |
| **Decorrelate sex and salary (S3):**
`SYNTHESIZE: Adult;`
`  ENFORCE: STATISTICAL:`
`    (E[sex * salary] - E[sex] * E[salary])`
`     / (STD[sex] * STD[salary] + 0.00001) == 0;`
`END;` | 0.7525 |

marginals that are to be preserved well by the final model) AIM works by iterating the following steps: (i) using the exponential mechanism to select a marginal from the workload to be measured, (ii) privately measuring this selected marginal using the Gaussian mechanism, (iii) fitting a generative model to all the privately measured marginals up to this point, and (iv) increasing the per-iteration budget $\epsilon_t$ in case the improvement obtained from the new measurement is insufficient. The steps (i)–(iv) are repeated until the entire privacy budget $\epsilon$ is used up. We adapt this algorithm by replacing the graphical model used in AIM with our generative model $g_\theta$ for step (iii) and training it similarly as in the non-private setting using the privately measured marginals as reference. Additionally, we modify step (iv) in a similar vein to adaptive ODE solvers by also allowing for a decrease in the budget in case the model showed a strong improvement in the given iteration, which we detail below.

Let $\gamma_t$ be the privacy parameter of the selection step (parameter of the exponential mechanism), and $\sigma_t$ the privacy parameter of measurement step (parameter of the Gaussian mechanism), each at iteration $t$. Also, let the sample generated by $g_\theta$ at iteration $t$ be denoted as $X_t$, and denote the marginal of the features $r$ selected in round $t$ measured on the sample $X_t$ with domain size $n_r$ as $M_r(X_t)$. Then, using the budget annealing step of AIM, one doubles $\gamma_t$ and halves $\sigma_t$ if $\|M_r(X_t) - M_r(X_{t-1})\|_1 \leq \sqrt{2/\pi} \cdot \sigma_t \cdot n_r$, *i.e.,* the per-round privacy budget is increased $4\times$ whenever the change in marginals is smaller than the expected error at the current noise level. Although this choice is well motivated by McKenna et al. (2022), we found that for ProgSyn this led to too few rounds of private training, as the per-round budget is only increased, and never decreased, when for example the improvement at the current round was much better than expected. Especially, as the increase every time is 4-fold, the budget was depleted very quickly, leading to poor results. Therefore, we modified this annealing step in AIM by (i) allowing for a decrease in the per-round budget in case the measurement provided an improvement that is larger than the expected error, and (ii) setting a maximum adaptation factor of $\sqrt{2}$, meaning that the per-round privacy budget changes at most $2\times$ in each round. Our new annealing step is shown in Algorithm 1.

---

**Algorithm 1** ProgSyn Privacy Budget Annealing

---

1: $\xi \leftarrow \dfrac{\|M(X_t) - M(X_{t-1})\|_1}{\sqrt{2/\pi} \cdot \sigma_t \cdot n_r}$
2: **if** $\xi \leq 1$ **then**
3: $\quad \sigma_{t+1} \leftarrow \max(\xi, 1/\sqrt{2}) \cdot \sigma_t$
4: $\quad \gamma_{t+1} \leftarrow \gamma_t / \max(\xi, 1/\sqrt{2})$
5: **else**
6: $\quad \sigma_{t+1} \leftarrow \min(\xi, \sqrt{2}) \cdot \sigma_t$
7: $\quad \gamma_{t+1} \leftarrow \gamma_t / \min(\xi, \sqrt{2})$
8: **end if**

---

Table 19: Commands and hyperparameters used in the experiment:
**Logical Constraints** (non-private)

| Command | Weights |
|---|---|
| **Logical Implication (I1):**
```SYNTHESIZE: Adult;```
```  ENFORCE: IMPLICATION:```
```    marital_status == Widowed OR relationship == Wife```
```      IMPLIES sex == Female;```
```END;``` | 0.0000075 |
| **Logical Implication (I2):**
```SYNTHESIZE: Adult;```
```  ENFORCE: IMPLICATION:```
```    marital_status in {Divorced, Never_married}```
```      IMPLIES relationship not in {Husband, Wife};```
```END;``` | 0.0000075 |
| **Logical Implication (I3):**
```SYNTHESIZE: Adult;```
```  ENFORCE: IMPLICATION:```
```    workclass in {Federal_gov, Local_gov, State_gov}```
```      IMPLIES education in {Bachelors, Some_college,```
```       Masters, Doctorate};```
```END;``` | 0.0000075 |
| **Logical Row Constraint (R1):**
```SYNTHESIZE: Adult;```
```  ENFORCE: LINE CONSTRAINT:```
```    sex == Female;```
```END;``` | 0.0000025 |
| **Logical Row Constraint (R2):**
```SYNTHESIZE: Adult;```
```  ENFORCE: LINE CONSTRAINT:```
```    age > 35 AND age < 55;```
```END;``` | 0.0000075 |
| **Combined Command:**
```SYNTHESIZE: Adult;```
```  ENFORCE: IMPLICATION:```
```    marital_status == Widowed OR relationship == Wife```
```      IMPLIES sex == Female;``` | 0.0000075 |
| ```  ENFORCE: IMPLICATION:```
```    marital_status in {Divorced, Never_married}```
```      IMPLIES relationship not in {Husband, Wife};``` | 0.0000075 |
| ```  ENFORCE: IMPLICATION:```
```    workclass in {Federal_gov, Local_gov, State_gov}```
```      IMPLIES education in {Bachelors, Some_college,```
```       Masters, Doctorate};``` | 0.0000075 |
| ```  ENFORCE: LINE CONSTRAINT:```
```    sex == Female;``` | 0.0000025 |
| ```  ENFORCE: LINE CONSTRAINT:```
```    age > 35 AND age < 55;```
```END;``` | 0.0000075 |

Table 20: Commands and hyperparameters used in the experiment:
**Logical Constraints** (private)

| Command | Weights |
|---|---|
| **Logical Implication (I1):**
```SYNTHESIZE: Adult;```
`  ENSURE: DIFFERENTIAL PRIVACY:`
`    EPSILON=1.0, DELTA=1e-9;`
`  ENFORCE: IMPLICATION:`
`    marital_status == Widowed OR relationship == Wife`
`     IMPLIES sex == Female;`
`END;` | 0.00005 |
| **Logical Implication (I2):**
`SYNTHESIZE: Adult;`
`  ENSURE: DIFFERENTIAL PRIVACY:`
`    EPSILON=1.0, DELTA=1e-9;`
`  ENFORCE: IMPLICATION:`
`    marital_status in {Divorced, Never_married}`
`     IMPLIES relationship not in {Husband, Wife};`
`END;` | 0.0000125 |
| **Logical Implication (I3):**
`SYNTHESIZE: Adult;`
`  ENSURE: DIFFERENTIAL PRIVACY:`
`    EPSILON=1.0, DELTA=1e-9;`
`  ENFORCE: IMPLICATION:`
`    workclass in {Federal_gov, Local_gov, State_gov}`
`     IMPLIES education in {Bachelors, Some_college,`
`      Masters, Doctorate};`
`END;` | 0.000375 |
| **Logical Row Constraint (R1):**
`SYNTHESIZE: Adult;`
`  ENSURE: DIFFERENTIAL PRIVACY:`
`    EPSILON=1.0, DELTA=1e-9;`
`  ENFORCE: LINE CONSTRAINT:`
`    sex == Female;`
`END;` | 0.0000375 |
| **Logical Row Constraint (R2):**
`SYNTHESIZE: Adult;`
`  ENSURE: DIFFERENTIAL PRIVACY:`
`    EPSILON=1.0, DELTA=1e-9;`
`  ENFORCE: LINE CONSTRAINT:`
`    age > 35 AND age < 55;`
`END;` | 0.0000125 |

Table 21: Commands and hyperparameters used in the experiment:
**Stacking Constraints of Different Types**

| Command | Weights |
|---|---|
| **Full Program:** | |
| `SYNTHESIZE: Adult;` | |
| `  MINIMIZE: FAIRNESS:` | |
| `    DEMOGRAPHIC_PARITY(protected=sex, target=salary,` | 0.0009 |
| `      lr=0.1, n_epochs=15, batch_size=256);` | |
| `  ENFORCE: STATISTICAL:` | |
| `    E[age] == 30;` | 0.000025 |
| `  ENFORCE: STATISTICAL:` | |
| `    E[age|sex==Male] == E[age|sex==Female];` | 0.0000125 |
| `  ENFORCE: IMPLICATION:` | |
| `    workclass in {Federal_gov, Local_gov, State_gov}` | 0.0000075 |
| `      IMPLIES education in {Bachelors, Some_college,` | |
| `        Masters, Doctorate};` | |
| `  ENFORCE: IMPLICATION:` | |
| `    marital_status in {Divorced, Never_married}` | 0.0000075 |
| `      IMPLIES relationship not in {Husband, Wife};` | |
| `END;` | |

## F   DIFFERENCES TO GEM

The main difference to the fixed-noise model used in Liu et al. (2021) (GEM) is that ProgSyn resamples the input noise at each training step, and therefore, it truly learns a generative model of the data with respect to the Gaussian distribution at the input. Additionally, our final layer is different from the one used in GEM, where the authors use a simple per-feature softmax head and conduct the training of the network in a relaxed representation space. Only once the training is done, for generating a final sample, the authors of GEM project the output of their model to the correct one-hot representations by sampling each feature independently in proportion to their obtained values in the relaxed representation. Whereas, we use a straight through estimator Gumbel Softmax (Jang et al., 2017) at the output, meaning that we already conduct the training in the hard, one-hot encoded space. Finally, for DP training we use a modified version of the selection and privacy budgeting algorithm presented in McKenna et al. (2022), and not the method presented in Liu et al. (2021). We explained the modifications we conduct in Appendix E.

## G   BROADER IMPACT

ProgSyn is the first programmable synthetic data generation method for tabular data, opening the possibilities of data sharing in areas where previously this was limited, due to privacy, bias, or proprietary issues. As such, we hope that ProgSyn, can open the way towards democratizing access to data, by allowing even entities that were previously reluctant to share their records to publish them. Such a development would be beneficial not only for the open-source community, and the science community but also for the industrial players providing the data, who could benefit from the open-sourced developments using their data themselves as well.

However, we have to acknowledge that allowing for mechanical manipulations in the data could open the door for malicious actors to purposefully modify their data releases in a misleading or straight-out harmful way. Although one could argue that ProgSyn makes this process potentially easier, our contribution is still significant from a mitigation perspective. We raise awareness that such manipulations on the data are possible and appeal to future work to approach this issue either from the technical or the legal end.

## H   LIMITATIONS

Although ProgSyn achieves competitive results in synthetic data generation, there are some design choices that may limit its performance. One of these factors is the fact that the data has to be discretized prior to fitting ProgSyn, where we only used a simple uniform discretization scheme with 32 buckets. We made this choice as the goal of this paper was not to present the best pure

synthetic data generation method, and we believe that a more carefully chosen discretization scheme could improve the inherent performance of ProgSyn. Another delicate choice is the set of measured marginals for training, where, for simplicity, we resorted to three-way marginals without exploring other options. We believe that ProgSyn could greatly benefit from an advanced scheme choosing the marginals for training, which improvement is likely to translate into the constrained setting as well. In general, as ProgSyn relies on a generative model to learn the unconstrained data distribution, its intrinsic performance influences the quality of the constrained data. Therefore, we believe that any improvements to the generative model would, at least partially, translate into higher data quality in the constrained setting as well. Additionally, we continue to work on ProgSyn to incorporate a larger class of constraints, for example, constraints between pairs of generated data points.

