# OpenReview forum: "Programmable Synthetic Data Generation"
_ICLR.cc/2024/Conference — Submitted to ICLR 2024_

### Official Review · Reviewer_keAe · 2023-10-18

**Soundness:** 3 good
**Presentation:** 3 good
**Contribution:** 2 fair
**Rating:** 5
**Confidence:** 4

**Summary:**

This paper proposes a tabular data generation method called ProgSyn where one can vary fairness, privacy, and logical constraints. The three constraints are relaxed into differentiable loss terms and used to fine tune a generative model. Experiments show that it is possible to generate synthetic data that satisfies compound constraints while maintaining high downstream accuracy.

**Strengths:**

* This is a timely work that addresses the important problem of configurable tabular data generation satisfying multiple constraints.
* The presentation is straightforward to understand.
* Experiments show promising results on generating data satisfying multiple constraints.

**Weaknesses:**

* The support for fairness seems limited. There are many fairness definitions in the literature beyond demographic parity including equalized odds, equal opportunity, predictive parity, equal error rates, individual fairness, and causal fairness, which seem to be ignored here. The proposed work would be much more interesting if it could also be configured for the other fairness measures as well. Supporting demographic parity only gives the impression that only the easiest fairness measure is supported, and there is no discussion on how to possibly extend the framework to other measures either.

* Emphasizing the programmability of ProgSyn sounds a bit exaggerated. For example, Figure 3 looks like a conventional config file instead of say a Python program. When using DP, a user always specify epsilon and delta, but this is not called programming.

* It is not clear why DP should be optimized together with fairness holistically. Why not generate a fair dataset using an existing technique or a fairness-only version of ProgSyn and then add random noise to satisfy DP? This two-step approach should be a baseline and compared with ProgSyn empirically.

* It would be more interesting to see the limitations of this method where the accuracy actually has to drop in order to satisfy various constraints. The current experiments only show success cases, but fairness and privacy are not necessary aligning, so there has to be a point where the accuracy cannot be maintained. In Table 3, there is almost no reduction of accuracy after applying DP, which suggests that the proposed method may have not been stressed tested enough. Hence, there should be more extensive experiments showing what happens in truly challenging scenarios.

**Questions:**

Please address the weak points.

---

> ### Author Response · Authors · 2023-11-16
> **Rebuttal**
>
> First of all, we would like to thank the reviewer for their detailed assessment of our paper, and for their constructive feedback. We are also pleased to read the reviewer’s recognition of the timeliness and importance of the problem setting, the quality of presentation, and the strong empirical results. Below, we address the reviewers comments and questions:
>
> **Q1: Can ProgSyn support fairness notions beyond demographic parity?**
>
> Yes, there is no technical restriction in ProgSyn that would only allow for demographic parity, and the modularity of ProgSyn allowed us to easily implement and test two more fairness measures: equality of opportunity (EoO), and equalized odds (EO). We repeat the fairness experiment on Adult on these two new measures in all settings and report our results in the tables below:
>
> Equalized Odds (non-private) – True data acc.: 85.4% EO: 0.08
>
> | |Accuracy [%]|EO|
> |:--|:--:|--:|
> |Decaf DP|66.8|0.07|
> |Decaf FTU|69.0|0.14|
> |Decaf CF|67.1|0.08|
> |TabFairGAN|82.6|0.04|
> |ProgSyn|**84.5**|**0.031**|
>
> Equalized Odds (differentially private):
>
> | |Accuracy [%]|EO|
> |:--|:--:|--:|
> |PreFair Greedy|80.2|**0.01**|
> |PreFair Optimal|75.7|0.03|
> |ProgSyn|**83.4**|0.02|
>
> Equality of Opportunity (non-private) – True data acc.: 85.4% EoO: 0.09
>
> | |Accuracy [%]|EoO|
> |:--|:--:|--:|
> |Decaf DP|66.8|0.07|
> |Decaf FTU|69.0|0.15|
> |Decaf CF|67.1|0.10|
> |TabFairGAN|82.6|**0.02**|
> |ProgSyn|**84.5**|**0.02**|
>
> Equality of Opportunity (differentially private):
>
> | |Accuracy [%]|EoO|
> |:--|:--:|--:|
> |PreFair Greedy|80.2|**0.02**|
> |PreFair Optimal|75.7|0.04|
> |ProgSyn|**83.3**|0.04|
>
> As we can observe from the above tables, ProgSyn still prevails as the method providing the best accuracy at a low bias level. In the non-private setting ProgSyn achieves both the best fairness and accuracy from all methods, while in the differentially private setting, it achieves comparably low bias to PreFair while having >3% higher accuracy.
>
> In general, any fairness measure (as well as any other specification) that can be expressed as a continuous function or a good enough relaxation is naturally supported by ProgSyn. Therefore, there is no fundamental reason why individual fairness notions could not be integrated into ProgSyn either, however, we do not consider it here as (i) there are conflicting notions of individual fairness, (ii) the design of individual fairness pipelines involve challenges that are orthogonal to the objective of our work, and (iii) the fair synthetic tabular data generation methods we have surveyed all concern group fairness, which is also where we position our work. We definitely believe that developing a flexible individually fair synthetic data generation pipeline is an interesting research challenge that could lead to highly valuable contributions for the community.
>
> Further, regarding causal fairness, we did not consider this fairness notion as it requires knowledge of the causal structure of the dataset, introducing a tight, second information bottleneck in the pipeline, and additional challenges worth their own contributions, such as the question of how can the elucidation of the causal structure be united with differential privacy. The requirement for the causal graph is a serious limitation of causal fairness methods, as also exemplified by DECAF, one of the fair synthetic data generation methods we compare ProgSyn against, where we were only able to conduct comparisons on Adult, and not on the other three datasets, as the method’s implementation included only the causal graph of Adult.

---

> > ### Author Response · Authors · 2023-11-16
> > **Rebuttal**
> >
> > **Q2: Is the current approach to fairness under differential privacy constraints necessary for strong performance?**
> >
> > Yes, we believe so. Note also that, strictly speaking, we do not optimize fairness and differential privacy together in the current approach. Rather, we first conduct differentially private pre-training and then fine-tune for fairness, where the differential privacy guarantee is maintained due to the pos-processing property. As established by [1], DP can have a detrimental impact on the fairness of a given dataset, therefore it is necessary that the DP model/data is adapted for fairness achieving (or reinstating) a state of low bias.
> >
> > To demonstrate the weakness of a pipeline with an alternative ordering, we conduct the following experiment on Adult. We generate non-private fair synthetic data using ProgSyn (as it is the best current method for this purpose), then use the marginals obtained from these fair samples to train a differentially private ProgSyn. We compare the accuracy and the demographic parity distance w.r.t. to sex to the original pipeline in the table below:
> >
> > | |Accuracy [%]|Dem. Par.|
> > |:--|:---:|:---:|
> > |First DP then Fairness (as in the paper)| **82.1**|**0.01**|
> > |First Fairness then DP|81.6|0.05|
> >
> > As we can observe, the currently employed pipeline prevails both in terms of accuracy and bias score.
> >
> > **Q3: Can you please discuss the performance of ProgSyn under several simultaneous constraints in more detail? At which point does it break down?**
> >
> > Yes, we appreciate the reviewer's question, and we also believe that it can be informative to explore the extremes of the method. Our original consideration with the experiment was to include a reasonable amount of constraints, i.e., a number that could commonly come up in usual use-cases, and see if the method can still perform at a satisfactory level.
> >
> > Note that we still observe an accuracy trade-off for added constraints, however, it is rather low, with the method qualitatively maintaining roughly the same performance level. Smoother decreases, but still within reasonable minimal performance, can be observed on the same experiment on other datasets, as shown in Tables 10-12 in the appendix.
> >
> > To complete the picture, we added more constraints, even contradicting constraints to the experiment on Adult, and reordered it such that we can observe a smoother decrease of the accuracy:
> >
> > |Acc. [%]|Mean Age|M-F Age Diff|I3|I2|Cov(sex, y)|R2|I1|Dem. Par.|
> > |:---:|:---:|:---:|:---:|:---:|:---:|:---:|:---:|:---:|
> > |85.1|37.29|2.33|0.60|0.99|0.22|0.40|0.93|0.19|
> > |84.5|30.20|1.46|0.59|0.99|0.22|0.34|0.93|0.20|
> > |84.5|30.18|0.02|0.59|0.99|0.22|0.33|0.93|0.20|
> > |84.7|30.18|0.03|0.99|0.98|0.22|0.34|0.93|0.18|
> > |84.6|30.19|0.04|0.98|1.00|0.22|0.34|0.92|0.19|
> > |83.0|30.19|0.00|0.93|1.00|0.00|0.33|0.92|0.11|
> > |81.3|35.28|0.02|0.98|1.00|0.00|1.00|0.93|0.04|
> > |80.3|35.28|0.01|0.92|1.00|0.00|1.00|1.00|0.04|
> > |77.5|35.28|0.01|0.96|1.00|0.00|1.00|1.00|0.11|
> >
> > For the sake of illustration, we did not employ rejection sampling over the logical constraints here, such that we have a better insight in the influence of the constraints on eachother. As we can see, with this many complex constraints the performance smoothly decreases as more and more constraints are added, to the point where the satisfaction scores of certain specifications are starting to worsen as well. The breaking point happens, when the sixth constraint R2 is introduced, which constrains the age feature to be between 35 and 55, which contradicts the first constraint that is setting the mean age to 30 and also interferes with the second constraint which is trying to equalize the mean age of males and females. This destabilizes the optimization, and a larger than expected accuracy loss is incurred. From this point on, additional constraints put further strain on the model, with the accuracy decreasing until 77.5%. Nevertheless, even at this complexity level, the model still produces data that is compliant with most of the given specifications.
> >
> > Note that the above breakdown is lemon-picked, and we believe that most use-cases will not include logically contradicting and several interfering specifications.

---

> > > ### Author Response · Authors · 2023-11-16
> > > **Rebuttal**
> > >
> > > **Q4: Does the accompanying domain specific language (DSL) provide further programmability than that achievable by simple configuration files?**
> > >
> > > Yes, which we believe is also clearly demonstrated by the examples provided both in the main paper and in the list of commands used in the appendix. Note that we support composite expressions both for logical and statistical constraints that naturally require recursive parsing, best facilitated in a DSL. Note also that programmability here does not refer to a general purpose programming language, but rather to an easy-to-use and intuitive interface in which the complex specifications can be declared in a fashion that closely resembles those expressions' mathematical definition.
> > >
> > > To exemplify the usefulness of ProgSyn’s programmability, consider the following example on how the user has to input the constraint I2 from Table 2 to ProgSyn, SDV models, and to AIM, respectively:
> > >
> > > ProgSyn:
> > > ```
> > > ENFORCE: IMPLICATION:
> > > 	marital_status in {Divorced, Never_married} IMPLIES relationship not in {Husband, Wife}
> > > ```
> > >
> > > SDV:
> > > ```
> > > def is_valid_I2(column_names: list, data: pd.DataFrame, extra_parameter) -> pd.Series:
> > > 	data = data.reset_index(drop=True)
> > > 	validity_filter = np.ones(len(data)).astype(bool)
> > > 	antecedent_mask = (data['marital-status'] == 'Divorced') | (data['marital-status'] == 'Never-married')
> > > 	antecedent_indices = data[antecedent_mask].index.to_numpy()
> > > 	consequent_mask = np.logical_and((data[antecedent_mask]['relationship'] != 'Husband').to_numpy(), (data[antecedent_mask]['relationship'] != 'Wife').to_numpy())
> > > 	validity_filter[antecedent_indices] = consequent_mask
> > > 	return pd.Series(validity_filter)
> > > ```
> > >
> > > AIM:
> > > ```
> > > szeros = {('marital-status', 'relationship'): [(1, 2), (1, 0), (2, 2), (2, 0)]}
> > > ```
> > >
> > > As we can see, ProgSyn provides a compact, intuitive, and practical interface for declaring constraints.
> > >
> > > **References:**
> > >
> > > [1] G Ganev et al. Robin hood and matthew effects: Differential privacy has disparate impact on synthetic data. ICML 2022. https://proceedings.mlr.press/v162/ganev22a.html

---

> > > > ### Author Response · Authors · 2023-11-21
> > > > **Follow-Up**
> > > >
> > > > We belive to have addressed the reviewer's open concerns in our rebuttal, demonstrating the framework's support of fairness measures beyond demographic parity in Q1, underlining the effectiveness of the current fairness-DP pipeline comparing it to an alternative in Q2, providing a harder stress test on constraint chaining in Q3, and discussing the advantages of the programmable interface of ProgSyn in Q4. In case the reviewer has any remaining or follow-up questions, concerns, or comments, we are eager to engage until the end of the author/reviewer discussion phase on the 22nd Nov. AOE.

---

### Official Review · Reviewer_XJ8v · 2023-10-30

**Soundness:** 3 good
**Presentation:** 3 good
**Contribution:** 2 fair
**Rating:** 5
**Confidence:** 3

**Summary:**

This article introduces the ProgSyn framework, which is the first framework designed for programmable table data generation.  The overall architecture of this framework is based on a two-stage process, starting with pre-training of the generated model using sampling and decoder structures, and then fine-tuning specific downstream tasks and adapting to their requirements.  At the same time, in the process of concrete implementation, ProgSyn uses the relaxed version of differential privacy, descriptive requirements and specification to achieve the programmability of the whole process.  The structure of this paper is reasonable and the content is clear, which provides a good model for the research of controlled table data generation.  In subsequent experiments and appendices, the authors also intelligently present their contributions and ideas by using selected datasets.

**Strengths:**

1.	The paper maintains a clear and focused main theme, presenting novel research in the realm of programmable generation for table data. The proposed ProgSyn framework, utilizing a pretraining and fine-tuning architecture, effectively caters to the requirements of downstream tasks in practical scenarios.
2.	In terms of methodology, the authors employ three approaches: differential privacy constraints, logical constraints, and statistical formulations. These strategies support the programmable nature of ProgSyn and demonstrate careful consideration for its differentiability, thereby enhancing its practical implement ability.
3.	Within this paper, the proposed differential computation for binary masks addresses the challenge of non-differentiable hard logical constraints and counts. This method provides an effective means for controlling the content generated by the model during the generation process.
4.	The supporting materials provided in this paper exhibit well code writing standardization and good applicability. The code content aligns well with the paper's core ideas, making it an excellent resource for readers seeking a deeper understanding of the author's concepts.

**Weaknesses:**

1.	Theoretical Framework: Although the research area explored in this paper holds promise for further investigation, the technical aspects in the paper appear somewhat dated, primarily covering foundational theories and methods. Considering the goal of ProgSyn is to generate sufficiently realistic simulated data with privacy protection properties, the authenticity aspect is addressed mainly in terms of experimental accuracy (given that XGBoost is a robust classifier), without validating its reliability from a statistical hypothesis testing perspective. This arrangement may lead to a somewhat one-sided argument in the paper.
2.	Writing Clarity: The content in the paper's introduction appears somewhat disorganized, as it combines background introduction with an overview of the framework's methodology. It is recommended to organize the sections logically as "introduction," "related work," and "formulation" to help readers better understand the core content of the paper. Additionally, there are instances of non-standard writing in the paper, such as lengthy formulas (page 6), pseudocode (page 5), page breaks (page 15, 20, 25), and inconsistencies in paper formatting (page 10-12). Also, it's important to address the improper use of color in tables.
3.	Code: To facilitate the research framework's wider adoption, it's advisable to update the code version requirements. Personally, I encountered issues with running the code on an NVIDIA GeForce RTX 4090 with CUDA capability sm_89, and it would be beneficial to address compatibility concerns to ensure broader accessibility and usability.

**Questions:**

Regarding this study's research, I have the following queries:
1.	Typically, table data is more widespread and common than image data. Does the controllable generation method proposed in this paper aim to fill the missing distributions in real data, as opposed to directly applying conditional filtering within the table? Because it is cheaper to filter tabular data through simple rules than to generate data compared to other data structures.
2.	In the field of image generation, we can rely on our own visual judgment or specific discriminators or metrics, such as FID, for authenticity assessment. Apart from the loss function control methods mentioned in this paper, are there more reliable approaches for verifying the authenticity of the generated data?
3.	In practical scenarios involving table data, recommendation systems present a more realistic application. Since relying solely on a highly generalizable model like XGBoost might not provide strong model performance validation, can ProgSyn consider further methodological validation (e.g., using models from other domains like ONN, xDeepFM, or more general models like SVM, GBDT)?
4.	In the main text, I didn't come across rigorous proofs related to statistical control of table data generation. Could you provide some information on the formulation process for statistical control? This would greatly assist readers in understanding controllable generation.

---

> ### Author Response · Authors · 2023-11-16
> **Rebuttal**
>
> First of all we would like to thank the reviewer for their thorough investigation of our work, including the supplementary materials, and for their far-reaching and constructive feedback. We also appreciate the reviewer’s recognition of the novelty of our work, its strong empirical performance, and the quality of the provided artifacts. Below we address your questions and points raised:
>
> **Q1: Does the empirical evaluation in the paper provide a compelling argument?**
>
> Yes, we believe so. The evaluation method used in the paper to obtain our results follows the practices established in the literature on synthetic tabular data generation (e.g., [1, 2, 3, 4]). Although we also believe that providing a robust and principled statistical hypothesis test for validating the synthetic data would be ideal, this is currently not the method of evaluation in this field, due to it being fundamentally hard. Due to the high complexity and dimensionality of the involved distributions, there is currently no gold-standard method that is tractable and reliable to a degree that would allow it to be adopted by the field of synthetic tabular data research for evaluation. The closest to this comes the comparison of low dimensional marginals, which is the basic building block of our method already. All in all, we believe that strong contributions can be made here that would benefit the community, but addressing this problem is beyond the scope of our work.
>
> In more detail, we evaluate mainly using the downstream accuracy of a state-of-the-art classifier, and, as shown in the experiments in Appendix C, we also measure the TV-distance between the learned and the true k-way marginals, effectively calculating a low-dimensional approximation of the divergence between the true and the modeled distribution. Note that we measured both metrics for all conducted experiments, merely, for presentational brevity, we opted to include the accuracy metric in the main paper, due to its easy understanding, and higher interest for practitioners. Furthermore, a strong classifier is also a meaningful proxy summarizing how well basic to very high order correlations have been preserved in the synthetic sample with respect to the original data.
>
> As for assessing the alignment of the synthetic data with the custom specifications, we directly measure their satisfaction rates/degrees on the generated synthetic samples, providing the closest metric of interest.
>
> **Q2: Would using different models to evaluate the quality of the data change the overall picture of ProgSyn’s strong performance compared to other methods?**
>
> No, and for the sake of demonstration, we have re-run the fairness experiment on Adult adding more classifiers. In the table below we compare the performance of ProgSyn to the next best non-private method TabFairGAN on the classifiers Logistic Regression (LogReg, linear model), SVM, Random Forest (RF, tree ensemble), XGBoost (XGB, gradient boosted tree-based, used in the paper), and CatBoost (CB, gradient boosted tree-based):
>
> |Classifier|ProgSyn [Acc. / Dem. Parity]|TabFairGAN [Acc. / Dem. Parity]|
> |:---|:--:|:--:|
> |LogReg|82.1 / 0.01|76.1 / 0.01|
> |SVM|80.0 / 0.04|76.0 / 0.00|
> |RF|81.4 / 0.01|78.8 / 0.02|
> |XGB|82.1 / 0.01|79.8 / 0.02|
> |CB|82.0 / 0.01|80.5 / 0.01|
>
> As we can see, the overall picture is unchanged: ProgSyn prevails as the best method in terms of fairness-accuracy trade-off. Note that for LogReg and SVM, TabFairGAN is unable to produce data that is suitable for training and therefore results in classifiers that largely only predict the majority class for every instance, irrespective of any feature (including the protected feature, therefore such classifiers appear as fair, however are not useful as they are only able to assign every instance to the same class).
>
> Further, note that [5] asserts that for evaluating synthetic tabular data, one should use a state-of-the-art classifier, as weaker models are unable to exploit the edge in distributional detail that better synthesization methods provide, painting a false picture.
>
> Finally, we do not test on recommendation models, as, again in line with prior work, the datasets used in this paper are classification datasets. Note that this is representative, especially in the context of our work, as none of the referenced methods in which ProgSyn is positioned examine recommendation datasets, but focus mainly on classification datasets (e.g., [1, 2, 3, 4, 5, 6, 7, 8, 9]). Also, as table 10 in [10] highlights, albeit in the context of federated learning, only 2 out of 61 tabular data applications concern recommendation systems.

---

> > ### Author Response · Authors · 2023-11-16
> > **Rebuttal**
> >
> > **Q3: Do the technical aspects of the paper also represent state-of-the-art?**
> >
> > Yes, we firmly believe so. Note that in our scenario, the main technical challenges lie with (i) introducing an effective constraint system over the fundamentally non-differentiable domain of tabular data, and (ii) uniting vastly different scenarios and aspects of synthetic tabular data generation, such as, e.g., differential privacy and fairness, in a single, general, technical, and high-performance pipeline. Addressing these challenges, we make several significant and novel technical contributions by introducing differentiable relaxation to a wide range of constraints, and by carefully joining and extending several aspects and frameworks of synthetic tabular data generation methods, such that it allows for strong performance even under various hard-to-join tasks, specifications, and scenarios.
> >
> > Further, [2] highlights that in differentially private synthetic data generation, approaches focusing their objective on the statistics of the data source (such as our method training on the marginals) often strongly outperform more complex methods adapted from other domains, such as GANs.
> >
> > **Q4: “Does the controllable generation method proposed in this paper aim to fill the missing distributions in real data, as opposed to directly applying conditional filtering within the table? Because it is cheaper to filter tabular data through simple rules than to generate data compared to other data structures.”**
> >
> > No, the goal of our method is to learn to generate new synthetic samples based on a source dataset. The distribution of these synthetic samples has to adhere to two often conflicting objectives: (i) has to resemble the real data distribution as closely as possible to preserve utility in replacement of the real data, and (ii) has to follow the customization set by the user, such as providing differential privacy guarantees, exhibiting decreased bias, and other custom manipulations facilitated by ProgSyn. Note also that post-hoc filtering of the data is a subideal approach, as (i) often, the motivation to create synthetic data is data scarcity, therefore we cannot afford to throw away data points, (ii) it is unclear how non-row-wise constraints, such as fairness, or statistical could be facilitated, (iii) and privacy aspects would remain unaddressed.
> >
> > In general, filling in missing values in a table is referred to as data imputation in the literature, and extending existing tabular datasets with additional, varied samples is data augmentation. Although both of these problems are important and exhibit their own specialized line of work, the problem considered in this paper, synthetic data generation, follows the more general goal of learning the distribution (or a manipulated distribution as in our case) of a source tabular dataset. Note that synthetic data can be used for both of the above tasks, but it is mainly motivated by other data limitations (also highlighted in section 1 of our paper):
> >
> > - lack of (high quality) data: only a few real samples are available, e.g., because the real experiments are very expensive
> > - regulatory considerations: storing, processing, passing on, or deriving products from the real data is forbidden by regulations (e.g., GDPR)
> > - privacy: the data has to be decoupled from identifiable individuals, providing formal guarantees (for instance by using differential privacy)
> > - bias: the real data may exhibit biases that have to be mitigated before a data release is made
> > - proprietary information: the real data may contain higher order proprietary information that is difficult to filter on the real data, so an altered distribution has to be learned before a data release can be made
> >
> > At this point we would like to again highlight that ProgSyn is the first framework supporting **all** of these objectives.

---

> ### Author Response · Authors · 2023-11-16
> **Rebuttal**
>
> **Q5: “Could you provide some information on the formulation process for statistical control?”**
>
> Yes, let us illustrate the technique using the following example:
>
> ```
> ENFORCE: STATISTICAL:
> 	E[2 * age - capital_gain | education == Doctorate] >= 500;
> ```
>
> The above specification consists of a conditional expectation in a comparison term. Our method first calculates the statistical operation and then combines them in the formula using t-norms and DL2 [11] primitives.
>
> Proceedingly, let us first calculate the term `E[2 * age - capital_gain | education == Doctorate]`: here, the two involved features are `age` and `capital_gain`, therefore $\mathcal{S} = \\{\\texttt{age}, \\texttt{capitalgain}\\}$ (following the notation in the paper). The operation is the expectation $E$ , the features are involved in the function $f(\mathcal{S}: x, y) = 2 * x - y$ , and the condition $\phi$ is `education == Doctorate`. First, we calculate the binary mask belonging to the condition following the technique described for logical constraints, obtaining $b_{\phi}(\hat{X})$. Then, we apply the mask to  the sample $\hat{X}\_{\phi} = b_{\phi}(\hat{X}) \odot \hat{X}$, zeroing out all the rows where the condition does not apply. Using $\hat{X}\_{\phi}$, we calculate the joint marginal of the `age` and `capital_gain` features, obtaining: $\bar{\mu}(\\texttt{age},\\texttt{capitalgain}, \hat{X}\_{\phi})$, which is the empirical estimate of the distribution $p_{\theta}(\\texttt{age}, \\texttt{capitalgain}|\phi)$. Now, we can insert the obtained distribution into the mathematical definition of the expectation operation and obtain the value for our first term:
>
> $$t\_1 = \sum_{\\texttt{age}}\sum_{\\texttt{captialgain}} (2 * \texttt{age} + \\texttt{captialgain})\, \bar{\mu}(\\texttt{age},\\texttt{capitalgain}, \hat{X}\_{\phi})$$
> Notice that due to the careful steps taken, this process is fully differentiable, even though it involves conditioning and arithmetic expressions in discrete features.
>
> Next, we are left with the expression $t_1 > 500$ which is a logical expression of reals, for which we have to construct a differentiable loss term. Here we use the primitives introduced in [11] to construct a differentiable loss term:
>
> $$\mathcal{L}\_{\\texttt{stat}}(\hat{X} \leftarrow g\_{\theta}(z)) = max(0, 500 - t\_1).$$
> Notice that the term is positive, and therefore adds a loss penalty only if $t_1$ is less than $500$, i.e., the desired constraint is not satisfied. Otherwise, when the constraint is met, the loss is zero.
>
> **Q6: Please clarify the organization of the introduction section, and address other comments regarding the formatting of certain elements in the paper.**
>
> We believe that a good presentation in a paper is extremely important, therefore, we appreciate any feedback and comments on it, and are eager to incorporate them, further improving the paper.
>
> **Introduction:** We believe that the introduction section of our paper follows the common pattern of how introductions are and have to be structured, at least what is commonly adopted in the machine learning literature. We first start by setting a general scene to our problem (first paragraph), then we introduce the background and most related work, highlighting the research gap our work will target (the two paragraphs connected to “Synthetic data”), finally, we briefly introduce our method, structured in a short technical introduction, a working example, and a summary of our results. For some examples on this structure, see [4], [7], [12], [13], and [14], all published papers at prestigious venues. Nevertheless, we are always looking to improve our paper, therefore, if the reviewer has any concrete suggestions for improving the introduction, we would be highly appreciative of that.
>
> **Lengthy formulas:** Although due to the space restrictions of the paper format it will be unavoidable to sometimes have inline math formulas, we thank the reviewer for raising our awareness on readability concerns, and have uploaded a revised version of the paper.
>
> **Pseudocode:** In general, we believe that including excerpts of code in the paper, especially if it is core to its message, is not non-standard writing. Additionally, the referenced code snippet is not just mere pseudocode, but an actual example of how specifications can be defined and passed to ProgSyn, written in a domain specific language constructed by us for this task. If this has been unclear from the presentation in the paper, we are eager to improve it, and would be very grateful if the reviewer could share their concrete suggestions with us.

---

> > ### Author Response · Authors · 2023-11-16
> > **Rebuttal**
> >
> > **Page breaks:** As the appendix contains a plethora of tables and other floating objects, to ensure that they are organized within the context and chapter they belong to and therefore enhancing readability and overview, we have included page breaks after certain sections. In our updated version of the paper we have tried to reduce their number while aiming to maintain structural coherence.
> >
> > **Citation formatting:** We believe that the reviewer commenting on the formatting on pages 10-12 means the formatting of the citations in the references. Note that we use bibliography and other style files provided by the conference, and extract the bibtex citations for the referenced papers from google scholar, conference pages, and arxiv. As these bibtex entries may contain varying levels of information, the resulting references also may differ, yet *still conform with citation guidelines*. Nevertheless, we have uploaded a revised version of the paper, where we have post-processed the citations as much as possible using a citation cleaning tool *rebiber*, to achieve more homogeneity in the included information.
> >
> > **“improper use of color in tables”:** We used color in the tables to help readability and for ease of understanding. If the reviewer has other suggestions, we welcome them.
> >
> > **Q7: Although the code provides an excellent resource for further understanding, some compatibility issues arose running on an NVIDIA GeForce RTX 4090.**
> >
> > We appreciate the reviewer's efforts in looking at and attempting to run our code. We believe that good code writing practices and reproducibility is core to good research, and are pleased that the reviewer has recognized our efforts in this regard, listing our code as a strength of our paper. Although we thoroughly tested our code, we acknowledge that errors can occasionally occur, particularly when porting to new systems. Would it be possible for the reviewer to share the specific errors encountered, ideally paste the error message and reproduction details? Given the reviewer's description of their system, we suspect the problem arose from a driver-CUDA-pyTorch installation compatibility issue, but we would appreciate more details, as such error diagnoses ultimately can contribute also to us improving our code and descriptions. Thank you.
> >
> > **References**
> >
> > [1] L Xu et al. Modeling Tabular data using Conditional GAN. NeurIPS 2019. https://arxiv.org/abs/1907.00503
> >
> > [2] Y Tao et al. Benchmarking Differentially Private Synthetic Data Generation Algorithms. Privacy-Preserving Artifical Intelligence Workshop @ AAAI 2022. https://arxiv.org/abs/2112.09238
> >
> > [3] T Liu et al. Iterative Methods for Private Synthetic Data: Unifying Framework and New Methods. NeurIPS 2021. https://arxiv.org/abs/2106.07153](https://arxiv.org/abs/2106.07153
> >
> > [4] V Borisov et al. Language Models are Realistic Tabular Data Generators. ICLR 2023. https://arxiv.org/abs/2210.06280
> >
> > [5] A Kotelnikov et al. TabDDPM: Modelling Tabular Data with Diffusion Models. ICML 2023. https://proceedings.mlr.press/v202/kotelnikov23a.html
> >
> > [6] A Rajabi and O O Garibay. TabFairGAN: Fair Tabular Data Generation with Generative Adversarial Networks. Machine Learning and Knowledge Extraction 2022. https://www.mdpi.com/2504-4990/4/2/22
> >
> > [7] B van Breugel et al. DECAF: Generating Fair Synthetic Data Using Causally-Aware Generative Networks. NeurIPS 2021. https://proceedings.neurips.cc/paper/2021/hash/ba9fab001f67381e56e410575874d967-Abstract.html
> >
> > [8] D Pujol et al. PreFair: Privately Generating Justifiably Fair Synthetic Data. Proceedings of the VLDB Endowment 2023. https://dl.acm.org/doi/10.14778/3583140.3583168
> >
> > [9] R McKenna et al. AIM: An Adaptive and Iterative Mechanism for Differentially Private Synthetic Data. Proceedings of the VLDB Endowment 2022. https://dl.acm.org/doi/abs/10.14778/3551793.3551817
> >
> > [10]  S K Lo et al. A Systematic Literature Review on Federated Machine Learning: From a Software Engineering Perspective. ACM Computing Surveys 2021. https://dl.acm.org/doi/10.1145/3450288
> >
> > [11] M Fischer et al. DL2: Training and Querying Neural Networks with Logic. ICML 2019. https://proceedings.mlr.press/v97/fischer19a.html
> >
> > [12] M Balunovic and M Vechev. Adversarial Training and Provable Defenses: Bridging the Gap. ICLR 2020. https://openreview.net/forum?id=SJxSDxrKDr
> >
> > [13] N Carlini et al. Extracting Training Data from Large Language Models. USENIX Security Symposium 2021. https://www.usenix.org/conference/usenixsecurity21/presentation/carlini-extracting
> >
> > [14] J Geiping et al. Inverting Gradients - How easy is it to break privacy in federated learning?. NeurIPS 2020. https://proceedings.neurips.cc/paper/2020/hash/c4ede56bbd98819ae6112b20ac6bf145-Abstract.html

---

> > > ### Author Response · Authors · 2023-11-21
> > > **Follow-Up**
> > >
> > > In our rebuttal, we believe to have addressed the reviewers concerns and comments, discussing the evaluation method used in Q1, comparing the evaluation across different downstream models in Q2, discussing the technical aspects of ProgSyn in Q3, clarifying the problem setting in Q4, explaining the technical details of statistical customization on an example in Q5, addressing the reviewer's comments about presentation in Q6, and inquiring about their encountered CUDA incompatibility issue in Q7. In case the reviewer has any remaining or follow-up questions, concerns, or comments, we are eager to engage until the end of the author/reviewer discussion phase on the 22nd Nov. AOE.

---

### Official Review · Reviewer_Jzns · 2023-11-05

**Soundness:** 3 good
**Presentation:** 3 good
**Contribution:** 3 good
**Rating:** 6
**Confidence:** 5

**Summary:**

The authors tackle the interesting problem of adding constraints to the synthetic data generation. They provide a framework where they consider both statistical and logical constraints arising from privacy, fairness and the domain. Experiments are conducted on real-datasets to showcase the benefits of their approach.

**Strengths:**

Overall:

The paper is well-written and easy to digest. The approach is simple and the experiments are convincing but the novelty factor is a bit missing.

Pros:

(a) The problem setup is very timely and relevant to the literature and the community. The framework solves a real issue of generating high-quality synthetic data with constraints.
(b) The experiments are extensive and showcase the framework in a wide variety of constraints while contrasting with the current state of the art approaches.

**Weaknesses:**

Cons:

(i) The main approach of fine tuning and adding differentiable constraints is relatively straightforward.
(ii) The approach is not adaptive to changing the constraint set and is not even discussed in the paper.

**Questions:**

1. How would you generate a variety of synthetic datasets with varying constraint specifications without retuning your model?


* On the Constrained Time-Series Generation Problem  https://openreview.net/forum?id=KTZttLZekHa

---

> ### Author Response · Authors · 2023-11-16
> **Rebuttal**
>
> We would like to thank the reviewer for their acknowledgment of our timely and strong empirical contribution and quality of writing, their informative feedback, the informative resource provided, and their favorable overall assessment of the paper. Below we address their questions and points raised:
>
> **Q1: What novel technical and conceptual contributions does ProgSyn make?**
>
> Additionally to ProgSyn’s empirically demonstrated strong performance and high level of versatility, we firmly believe that ProgSyn brings both significant technical and conceptual novel contributions to the field of synthetic tabular data generation.
>
> Conceptually, it is the first work to facilitate extended customizability of the generated data, allowing for row-wise, distributional, and downstream-dependent specifications. Our framework cannot only be used to improve the synthetic data quality, but also to hand-tailor its distribution and other properties to specific, custom use-cases defined by the user. At the same time, ProgSyn effectively captures the capabilities of prior works across different aspects of tabular data generation.
>
> Technically, ProgSyn introduces several novel elements in the differentiable computation of the specification loss-terms, tackling several difficult challenges on the way. The common key challenge of each specification type arises from the fundamentally non-differentiable nature of categorical features in the produced tables. Once the conversion from complex constraints involving discrete features is achieved, using them as a regularizer in a two-staged optimization pipeline is indeed a natural approach for manipulating the modeled distribution to our liking. Nevertheless, note that ProgSyn makes its key technical contributions addressing the challenge of arriving at these differentiable loss terms (something that is for example in [1] in the domain of time series data is treated as a given). For instance, for logical constraints and conditions, we address this through our masking scheme introduced in Section 4. Further, we are the first to offer a technical solution to incorporate complex formulas involving conditional statistical expressions acting directly on the distribution of the data, going beyond row-wise constraints. We are also first to provide a general framework of incorporating downstream classifiers in the synthetic data generation training process; tackling the difficult additional challenges of training-in-the-loop, and uniting it with the rigorous requirements of differential privacy. Notably, this allowed us to set a new state-of-the-art on fair synthetic data generation, both in the non-private and the differentially private setting, while this objective not being the main focus of our work.

---

> > ### Author Response · Authors · 2023-11-16
> > **Rebuttal**
> >
> > **Q2: Discuss varying the constraints after the model has already been tuned for a different set of constraints.**
> >
> > Yes, in principle adaptation to a new constraint set without tuning is possible, although it would be ill-advised. For constraints that apply row-wise (like logical constraints), rejection sampling can be used, however, it will come with the limitations already discussed in the paper. For the rest of the constraint types it is also possible, as the model is differentiable with respect to the input noise as well, so one may optimize the noise at sampling time to adapt to new constraints.
> >
> > However, it is ill-advised, due to several factors: (i) the number of parameters in the noise is orders of magnitudes larger than the number of parameters in the model, making such optimizations slow and memory costly, and (ii) one of the most important features of ProgSyn are constraints that act on a distributional level and **not** on a row level, therefore ruling out the possibility of any guiding method relying on judging single samples at generation time being sufficient (such as for example the method used in [1], linked by the reviewer).
> >
> > Nevertheless, we believe that in the domain of ProgSyn there is **no need** for tuning-free adaptation, as the framework and as such the tuning, is extremely light-weight.
> >
> > Nevertheless, as we have demonstrated in Q2, as the framework is light-weight, effective adaptation can be conducted in just a few seconds, which, in our eyes, addresses the concerns coming with the necessity of potential frequent needs of adaptation. For instance, to adapt to the constraint R1 listed in Table 2, at the current parameter setting on Adult, we required around 3.5 minutes of fine-tuning at the employed hyperparameter settings.
> >
> > As we did not see this as a concern during development, we did not make further optimizations. However, note that adaptation time can be linearly reduced by: (i) reducing the batch size, (ii) increasing the chunking size at marginal calculation, or (iii) decreasing the number of epochs. For the sake of the argument, we repeated the mentioned experiment optimizing for fine-tuning time, and achieve **84.7% accuracy at 100% CSR** (equal to what is reported in the paper) in a mere **13.7 seconds** of fine-tuning resulting in a **15.8x speedup**, by only adjusting three hyperparameters in favor of execution speed: reducing the number of epochs to 100, reducing the batch-size to 2000, and increasing the chunk-size to 2000.
> >
> > Therefore, we believe that re-tuning the model in order to adapt it to new constraints is feasible.
> >
> > **References:**
> >
> > [1] A Coletta et al. On the Constrained Time-Series Generation Problem. To appear at NeurIPS 2024. https://arxiv.org/abs/2307.01717

---

> > > ### Author Response · Authors · 2023-11-21
> > > **Follow-Up**
> > >
> > > We belive to have addressed the reviewer's open concerns in our rebuttal, discussing the technical contributions of ProgSyn in Q1, and the variation of constraints after ProgSyn has been fitted to a fixed constraint set in Q2. In case the reviewer has any remaining or follow-up questions, concerns, or comments, we are eager to engage until the end of the author/reviewer discussion phase on the 22nd Nov. AOE.

---

### Author Response · Authors · 2023-11-16
**General Response**

We would like to thank all reviewers for their time spent reviewing our paper, and for their extensive and valuable assessment, comments, and feedback. We especially appreciate reviewers' comments deeming our work “interesting”, “effective”, “timely”, and “relevant”, while also recognizing our strong experimental results and quality of presentation and writing.

We address the reviewers’ questions, comments, and concerns in individual responses below.

Further, as part of this rebuttal, we have updated the manuscript, making the following changes:
- added a more detailed technical description of the improvements we have made over the AIM framework in differentially private training (Appendix E)
- added results on the Compas, and German Credit datasets on unconstrained generation (Appendix C)
- fixes of minor typos and writing details

---

### Author Response · Authors · 2023-11-21
**Author/Reviewer Discussion Phase Ending Soon**

We would like to kindly remind the reviewers that the author/reviewer discussion phase ends soon on the 22nd of Nov. AOE. We are still hopeful to engage in a fruitful discussion of our rebuttal.

---

### Meta-Review · Area_Chair_6BC6 · 2023-12-06

**Metareview:**

This paper proposes a generative model for tabular data that can satisfy declared statistical and logical expressions such as differential privacy or fairness. The proposed method represents all features in a one-hot format, matching the marginals of the model and training data, and uses a differential loss that penalizes violations of logical expressions. Three reviewers have reviewed the paper, acknowledging the necessity and relevance of the proposed model. However, two reviewers suggest rejection due to some reasons such as lack of writing clarity and insufficient validation of the proposed method's performance in a relatively new setting. This AC also reviewed the paper carefully and aligned with the negative opinions of the two reviewers for the following reasons:

- Continual variables must be converted into categorical variables and one-hot encoding must be used, which may limit the applicability of this paper as such transformations can lead to performance degradation in some datasets.
- The superiority of the marginal matching model has not been thoroughly verified, and applying other recent training methodologies for generative models, such as adversarial training (in GANs) or diffusion, is not possible.
- The evaluation of various components of the proposed method, such as handling logical constraints through regularization, rejection sampling, downstream specification, and their importance compared to baselines, is not adequately fair. For example, rejection sampling can be seamlessly applied to other baselines to force the logical constraint; this kind of trivial improvement for other baselines should be fairly considered, too. The pure effect of ProSyn in experiments like table 2 without rejection sampling has not been adequately verified etc.
- Lastly, the authors use xgboost accuracy to evaluate the proposed method's ability to force logical constraints, but the evidence for whether this is sufficient for evaluating how well it enforces logical constraints is lacking.

**Justification For Why Not Higher Score:**

evaluation for the proposed method is not sufficient.

**Justification For Why Not Lower Score:**

n/a

---

### Decision · Program_Chairs · 2024-01-16

Reject